# The time derivative of the geomagnetic field has a short memory

Mirjam Kellinsalmi[1,2], Ari Viljanen[1], Liisa Juusola[1], and Sebastian Käki[1,2]

[1]Finnish Meteorological Institute, Helsinki, Finland
[2]University of Helsinki, Helsinki, Finland

**Correspondence:** Mirjam Kellinsalmi (mirjam.kellinsalmi@fmi.fi)

**Abstract.**

Solar eruptions and other types of space weather effects can pose a hazard to the high voltage power grids via geomagnetically induced currents (GIC). In worst cases, they can even cause large scale power outages. GIC are a complex phenomenon, closely related to the time derivative of the geomagnetic field. However, the behavior of the time derivative is chaotic and has proven to be tricky to predict. In our study, we look at the dynamics of the geomagnetic field during active space weather. We try to characterize the magnetic field behavior, to better understand the drivers behind strong GIC events. We use geomagnetic data from the IMAGE (International Monitor for Auroral Geomagnetic Effect) magnetometer network between 1996 and 2018. The measured geomagnetic field is primarily produced by currents in the ionosphere and magnetosphere and secondarily by currents in the conducting ground. We use the separated magnetic field in our analysis. The separation of the field means, that the measured magnetic field is computationally divided into external and internal parts corresponding to the ionospheric and telluric origin, respectively. We study the yearly directional distributions of the baseline subtracted, separated horizontal geomagnetic field, $\Delta\mathbf{H}$, and its time derivative, $d\Delta\mathbf{H}/dt$. The yearly distributions do not have a clear solar cycle dependency. The internal field distributions are more scattered than the external field. There are also clear, station specific differences in the distributions related to sharp conductivity contrasts between continental and ocean regions or to inland conductivity anomalies. One of our main findings is that the direction of $d\Delta\mathbf{H}/dt$ has a very short "reset time", around two minutes, but $\Delta\mathbf{H}$ does not have this kind of behavior. These results hold true even with less active space weather conditions. We conclude that this result gives insight into the time scale of ionospheric current systems, which are the primary driver behind the time derivative's behavior. It also emphasises a very short persistence of $d\Delta\mathbf{H}/dt$ compared to $\Delta\mathbf{H}$, and highlights the challenges in forecasting $d\Delta\mathbf{H}/dt$ (and GIC).

## 1 Introduction

Space weather, eventually produced by eruptive phenomena in the Sun, can have harmful effects on Earth via, for example, *geomagnetically induced currents* (GIC). Usually GIC are weak and harmless, but due to stormy space weather they can even cause large-scale power outages. For example, in March 1989, a geomagnetic storm caused a province wide blackout in Québec, Canada (Bolduc, 2002). More thorough descriptions of space weather effects are given by, e.g., Boteler et al. (1998); Wik et al. (2009); Pulkkinen et al. (2005).

Even though the phenomenon of GIC has been studied for decades, we still do not have a complete understanding of the physics behind GIC events due to their complexity. To eventually forecast GIC events, we first need to understand the magnetic field dynamics behind them. The magnetic field that we can measure on the Earth's surface is primarily produced by *ionospheric* and *magnetospheric currents*, and secondarily by currents induced in the conducting ground, the *telluric currents*. We can use computational separation to divide the measured magnetic field into two parts; one that is created by currents in the ionosphere and magnetosphere (*external part*) and another that is created by the induced currents in the Earth's crust and mantle (*internal part*).

GIC is driven by the ground electric fields. These fields are associated with the time derivative of the geomagnetic field, $d\Delta\mathbf{B}/dt$, via Faraday's induction law. This is why the time derivative, $d\Delta\mathbf{B}/dt$, can be used as a proxy for GIC (Viljanen et al., 2001). However, the behavior of the derivative is complex and has proven to be difficult to predict (Pulkkinen et al., 2011; Kwagala et al., 2020). Especially, it is a big challenge to produce accurately both the vector $d\Delta\mathbf{B}/dt$ (magnitude and direction) and the occurrence time of large $d\Delta\mathbf{B}/dt$.

Several studies have been done focusing on $d\Delta\mathbf{B}/dt$. The study by Viljanen et al. (2001) looks at the occurrence of large values of the ground horizontal $d\Delta\mathbf{B}/dt$ on a daily, seasonal, and yearly levels and their directional distributions at IMAGE magnetometer stations in northern Europe. One of the study's findings, regarding the directional distribution of $d\Delta\mathbf{H}/dt$ (the horizontal part of $d\Delta\mathbf{B}/dt$), is that there is no evident solar cycle dependence, but the distribution pattern is narrower in the quietest and most active years of the cycle. Viljanen and Tanskanen (2011) take a closer look on the diurnal and seasonal distributions of large $d\Delta\mathbf{H}/dt$. Among other things they find that large $d\Delta\mathbf{H}/dt$ occur most commonly around local MLT midnight and early morning hours, and very rarely around midday. Also, large $d\Delta\mathbf{H}/dt$ happen mainly during westward electrojets, with southward oriented $\mathbf{H}$. One of the main findings of Pulkkinen et al. (2006) is that there is a clear change in the dynamics of magnetic field fluctuations in temporal scale from 80 to 100 seconds. They conclude that above scales of 100 s, the spatiotemporal behavior of $d\Delta\mathbf{H}/dt$ resembles that of uncorrelated white noise. Juusola et al. (2020) found that the internal part, $d\Delta\mathbf{H}_{int}/dt$, is comparable to, or even larger than the external part, $d\Delta\mathbf{H}_{ext}/dt$. Their results also show that the directional distribution of $d\Delta\mathbf{H}_{int}/dt$ is much more complex than that of $d\Delta\mathbf{H}_{ext}/dt$, which is explained by the 3D ground conductivity and associated telluric currents.

Our group is approaching the problem of GIC from a slightly different perspective than previous studies. Many GIC studies based on the time derivative of the ground magnetic field, e.g., Pulkkinen et al. (2006); Viljanen et al. (2001); Viljanen and Tanskanen (2011), concentrated on the total $d\Delta\mathbf{H}/dt$, which is a sum of the external and internal contribution. Instead, we use separated magnetic field measurements to find indicators for strong GIC events. Our primary interest is to deepen previous understanding of the characteristics of the magnetic field and its time derivative during active events characterized by large values of $d\Delta\mathbf{H}/dt$. In this paper, we analyze both the external and internal part of $\Delta\mathbf{H}$ and $d\Delta\mathbf{H}/dt$ and study their temporal and spatial differences.

## 2 Data and methods

### 2.1 Data

We use 10 s data from the IMAGE (International Monitor for Auroral Geomagnetic Effects) magnetometer network between 1996-2018. Locations of the IMAGE magnetometers at the beginning of 2017 are presented in Fig. 1. Quiet-time baselines are subtracted from the data using an automatic method (van de Kamp, 2013).

In this study, we use magnetic data separated into external and internal parts, as was done by Juusola et al. (2020). We use the 2D Spherical Elementary Current System method (SECS) to perform the separation. In this method there are two layers of elementary currents used, one in the ionosphere (90 km altitude) and the other just below the ground (0 km, for numerical reasons set to 1 m). In our implementation of the 2D SECS method, the cutoff parameter for singular values of the singular values decomposition is zero. As a consequence, all components of the observed geomagnetic field are perfectly reproduced at all stations used in the analysis. A thorough description of the SECS method is given by Vanhamäki and Juusola (2020).

### 2.2 Methods

The measured, baseline subtracted, horizontal magnetic field vector is given as a time series, $\Delta \mathbf{H}(t)$, where $\Delta \mathbf{H} = \mathbf{H}_{measured} - \mathbf{H}_{baseline}$. Its direction is measured with respect to the (geographic) north direction positive clockwise ($\theta(t)$). See Fig. 2 for reference. We study the temporal change of $\theta$, i.e. $\Delta\theta$, as well as the relative change in the field amplitude, $R(T)$, over a time period, $T$. The parameter, $T$, is a multiple of the 10 s time step of the time series. $\Delta\theta$ is calculated for the total variation field ($\Delta \mathbf{H} = \Delta \mathbf{H}_{tot}$), external part ($\Delta \mathbf{H}_{ext}$) and internal part ($\Delta \mathbf{H}_{int}$). In the same way, we consider the time derivative ($d\Delta \mathbf{H}/dt$)

and the related direction. The relative change in the amplitude of the time derivative, is analyzed in a similar way. The main motivation behind this was to repeat a similar analysis, done in previous studies (e.g. Viljanen et al. (2001), Viljanen and Tanskanen (2011)) for the total field ($\mathbf{H}$), on the $\Delta \mathbf{H}_{ext}$ and $\Delta \mathbf{H}_{int}$.

The quantaties $\Delta \mathbf{H}$, $d\Delta \mathbf{H}/dt$, $\theta$, $\Delta\theta$, $R(T)$ and $T$ used in this study are defined in Table 1. Our study focuses on magnetic field behavior during active space weather, characterized by large values of $|d\Delta \mathbf{H}/dt|$. For the most cases, we use a threshold

value of $|d\Delta \mathbf{H}/dt| > 1$ nTs$^{-1}$, where $\Delta \mathbf{H}$ is the total, baseline subtracted, horizontal field. Since the used data is 10 s data, this limit value for the derivative means that the change in its amplitude is above 10 nT per 10 s. The specific questions we study are the following:

1. Is there yearly variation in directional distributions of $\Delta \mathbf{H}$ and $d\Delta \mathbf{H}/dt$?

2. How large is the geographic variability in these directional distributions and $\Delta\theta$?

3. Are there differences between the external and internal $\Delta \mathbf{H}$ and $d\Delta \mathbf{H}/dt$ in $\Delta\theta$?

4. What is the dependence of $\Delta\theta$ and $R(T)$ on $T$, and are there characteristic time scales?

5. Does the activity level, represented by $|d\Delta \mathbf{H}/dt|$, affect the directional and $\Delta\theta$ distribution?

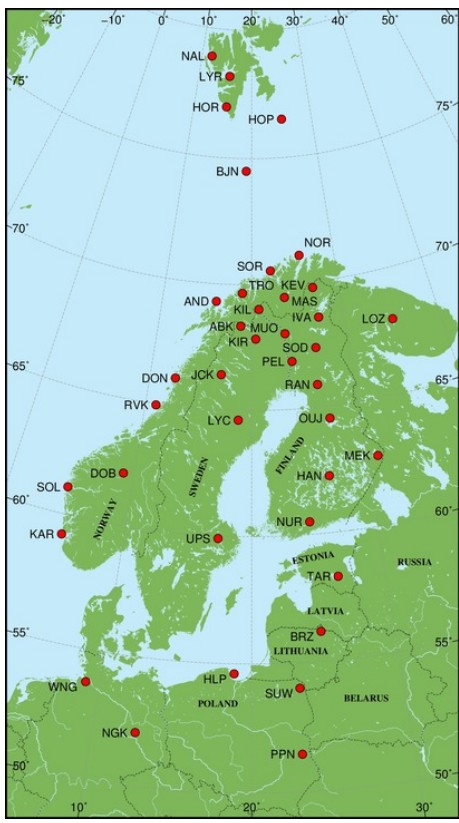

**Figure 1.** IMAGE station locations and name abbreviations in 2017 (IMAGE, 2021).

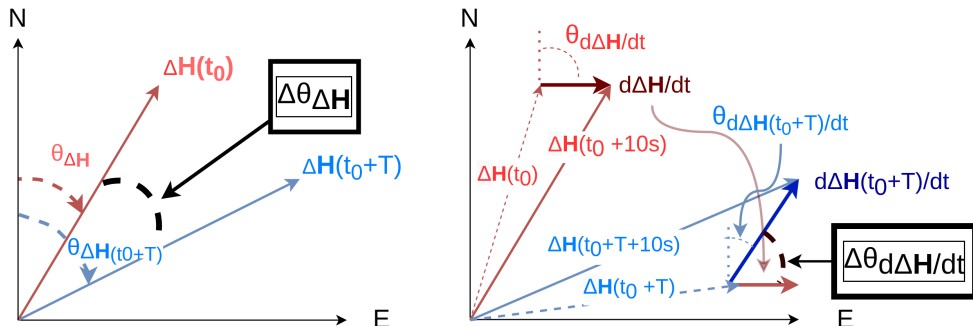

**Figure 2.** A schematic of the quantity $\Delta\theta$ for $\Delta\mathbf{H}$, and its time derivative, $\mathrm{d}\Delta\mathbf{H}/\mathrm{d}t$ used in this study. $\Delta\mathbf{H}(t)$ refers to the baseline subtracted magnetic field vector at a specific time, $t$. $\theta(t)$ refers to the angle between $\Delta\mathbf{H}(t)$ and the geographic north. $T$ is a multiple of the data sampling interval (10 s).

**Table 1.** Definitions for quantities used in this study. $\hat{\mathbf{e}}_x$ and $\hat{\mathbf{e}}_y$ are the northward and eastward unit vectors.

| | |
|---|---|
| Horizontal magnetic field vector | $\Delta\mathbf{H} = \Delta B_x\hat{\mathbf{e}}_x + \Delta B_y\hat{\mathbf{e}}_y$ |
| Amplitude | $\lvert\Delta\mathbf{H}\rvert = \sqrt{\Delta B_x^2 + \Delta B_y^2}$ |
| Horizontal magnetic field time derivative | $\mathrm{d}\Delta\mathbf{H}/\mathrm{d}t = \frac{\mathrm{d}\Delta B_x}{\mathrm{d}t}\hat{\mathbf{e}}_x + \frac{\mathrm{d}\Delta B_y}{\mathrm{d}t}\hat{\mathbf{e}}_y$ |
| Amplitude | $\lvert\mathrm{d}\Delta\mathbf{H}/\mathrm{d}t\rvert = \sqrt{\frac{\mathrm{d}\Delta B_x}{\mathrm{d}t}^2 + \frac{\mathrm{d}\Delta B_y}{\mathrm{d}t}^2}$ |
| Direction of the horizontal vector | $\theta = \arctan(\frac{\Delta B_y}{\Delta B_x})$ |
| Change in direction ($t_0$ = time when $\mathrm{d}\Delta\mathbf{H}/\mathrm{d}t$ reaches the threshold value) | $\Delta\theta = \theta(t_0 + T) - \theta(t_0)$ |
| Relative change in amplitude of $\mathrm{d}\Delta\mathbf{H}/\mathrm{d}t$ | $R(T) = \frac{\lvert\mathrm{d}\Delta\mathbf{H}/\mathrm{d}t\rvert_{t0+T}}{\lvert\mathrm{d}\Delta\mathbf{H}/\mathrm{d}t\rvert_{t0}}$ |
| Notation for the external and internal fields | $\Delta\mathbf{H}_{ext}$, $\Delta\mathbf{H}_{int}$, $\mathrm{d}\Delta\mathbf{H}_{ext}/\mathrm{d}t$ etc. |

We also look at the mean horizontal magnetic field directions at stations. Since we are dealing with circular data, we have to take additional measures to get a meaningful average direction. The directional distribution of the time derivative is bimodal, i.e., the values are clustered around two opposite directions (mainly north and south). The following method is used in the case of $\mathrm{d}\Delta\mathbf{H}/\mathrm{d}t$:

First we construct a histogram of eight bins of the directional values. The bins are: 1. $[0, 45)°$, 2. $[45, 90)°$, 3. $[90, 135)°$, 4. $[135, 180)°$, 5. $[180, 225)°$, 6. $[225, 270)°$, 7. $[270, 315)°$, 8. $[315, 360)°$. The second step is to find the highest bin, i.e. largest number of cases, which gives the approximate direction. (North: bins 1 and 8, East: 2 and 3, South: 4 and 5, West: 6 and 7.) The last step is to calculate the mean direction using only the values in the semicircle of the approximate direction. If the highest bin is in the east sector, calculate the mean direction using values in range 0°to 180°. For the sake of clarity, we present the mean direction in the case of the derivative for the south sector (90°to 270°) only. In other words, if the mean direction given by our method gave a northward direction, we add or subtract 180°. The method decribed here is a simple way to get an approximate mean direction for a circular, bimodal distribution. We also tried a few other methods (e.g. by Davis (2002)) for getting the mean direction, but they proved to be somewhat impractical with the very scattered distributions.

## 3  Results

### 3.1  Example event

We first look at the magnetic field behavior during a single space weather event. Figure 3 shows magnetic field data at Tromsø (TRO, geographic latitude = $69.66^o$ N) during one hour of the Halloween event in 2003. The panels, starting from the top, show the magnitude of the horizontal magnetic field ($\lvert\Delta\mathbf{H}\rvert$), $\Delta B_x$ and $\Delta B_y$ components, the magnitude of the time derivative of the field ($\mathrm{d}\Delta\mathbf{H}/\mathrm{d}t$), $\Delta\theta$ for $\Delta\mathbf{H}$, and $\Delta\theta$ for $\mathrm{d}\Delta\mathbf{H}/\mathrm{d}t$. The change in direction is calculated over $T$ = 1 min. The Halloween event was one of the strongest magnetic storms on record (Pulkkinen et al., 2005; Wik et al., 2009). $\lvert\mathrm{d}\Delta\mathbf{H}/\mathrm{d}t\rvert$ values are large ($> 10$ nTs$^{-1}$), indicating also strong GIC. We see that there is little variation in the direction of $\Delta\mathbf{H}$ (second lowest panel), whereas its time derivative (lowest panel) has much more chaotic behavior. $\mathrm{d}\Delta\mathbf{H}/\mathrm{d}t$ is changing direction very rapidly and

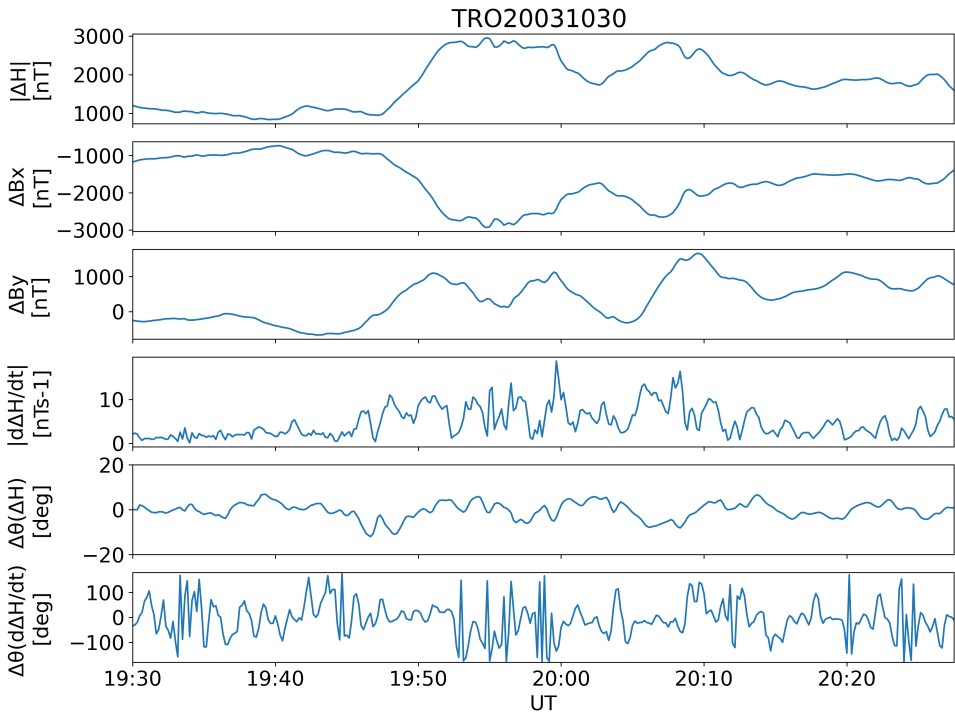

**Figure 3.** Different quantaties related to the horizontal magnetic field at Tromsø station during one hour of the Halloween event on 30th Oct 2003. Panels from the top are: 1) magnitude of the horizontal magnetic field, $\Delta\mathbf{H}$, 2) $\Delta B_x$ component, 3) $\Delta B_y$ component, 4) amplitude of the time derivative, $\mathrm{d}\Delta\mathbf{H}/\mathrm{d}t$, 5) change in direction, $\Delta\theta(T = 1\text{ min})$ of $\mathbf{H}$, 6) $\Delta\theta(T = 1\text{ min})$ of $\mathrm{d}\Delta\mathbf{H}/\mathrm{d}t$.

strongly during the whole period. We also point out that $|\Delta\mathbf{H}|$ remains steadily at a high level ($\sim 1000$ nT or larger) for tens of minutes, whereas $|\mathrm{d}\Delta\mathbf{H}/\mathrm{d}t|$ oscillates quickly between 0 and about 20 $\mathrm{nTs}^{-1}$. In other words, sequences of large $|\mathrm{d}\Delta\mathbf{H}/\mathrm{d}t|$ are short as also noted, for example, by Weygand et al. (2021, Fig. 2).

### 3.2   Location specific differences

Next, we examine directional distributions of the separated magnetic field at the IMAGE stations. Figure 4 shows polar plots

of the directional distributions of external and internal $\Delta\mathbf{H}$ at each station for one year (2017). The left panel shows $\Delta\mathbf{H}_{ext}$. We see very distinct southward distributions above latitude 64°. At lower latitudes the northward direction is dominant. The distributions are mostly narrow. As for $\Delta\mathbf{H}_{int}$, the right panel in Fig. 4, there seems to be more variation in directions. The behavior of the internal field is similar to that of the external one: southward orientations above 64°N, and northward (or very scattered) distributions below that latitude.

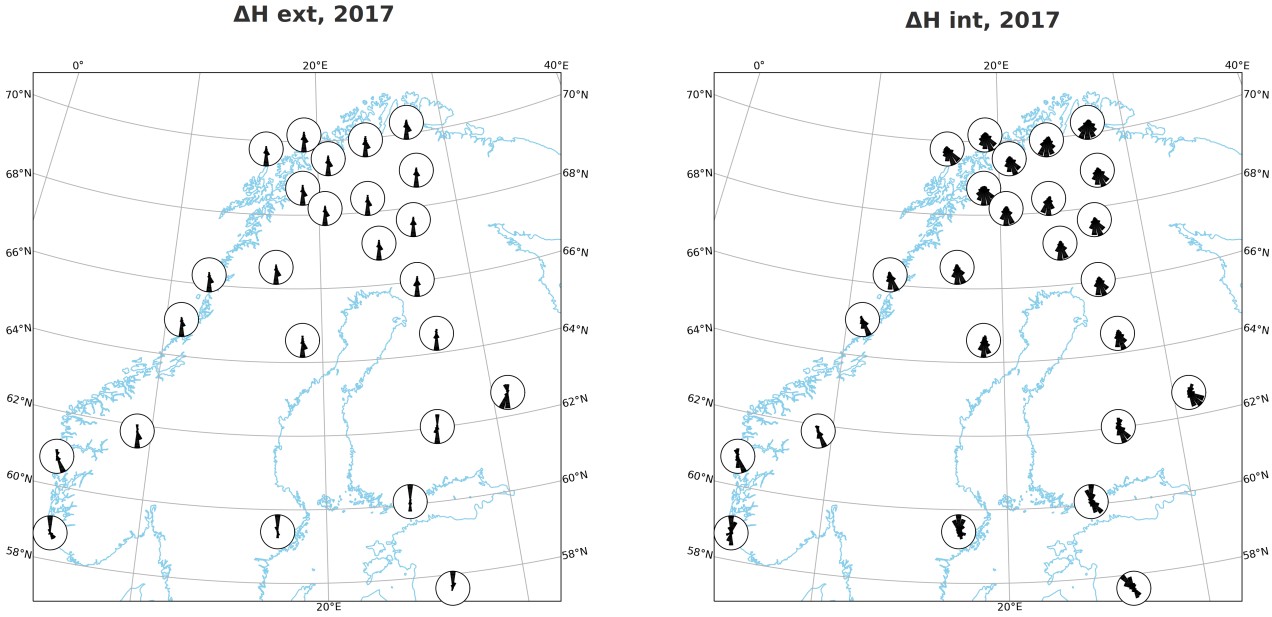

**Figure 4.** Directional distribution of external (left) and internal (right) $\Delta \mathbf{H}$ at IMAGE stations in 2017 when $|\mathrm{d}\Delta\mathbf{H}/\mathrm{d}t| > 1\mathrm{nTs}^{-1}$.

We repeat similar analysis on the time derivative of the external and internal field (Fig. 5). Left panel shows $\mathrm{d}\Delta\mathbf{H}_{ext}/\mathrm{d}t$ and right shows $\mathrm{d}\Delta\mathbf{H}_{int}/\mathrm{d}t$. The external field has, again, quite clear north-south orientations. There is a bit more scattering visible at the southern stations with less data.

As for the internal $\mathrm{d}\Delta\mathbf{H}/\mathrm{d}t$ there seems to be more variation between the stations. For example, Masi (MAS, geographic lat. = 69.46°N, lon. = 23.70°E) has a very clear north-east south-west orientation but in Tromsø (TRO, geographic lat. = 69.66°N, lon. = 18.94°E), the distribution looks almost even. Especially, some of the stations near the Norwegian coastline (e.g. DON, RVK) seem to have very narrow distributions.

The data from stations in Germany and Poland are available, but they were not included in these plots due to very limited amount of data points fitting the criterion ($|\mathrm{d}\Delta\mathbf{H}/\mathrm{d}t| > 1\ \mathrm{nTs}^{-1}$). The number of data points at each station in 2017 is presented in Table 2. As expected, the number of data points fitting the criterion increases towards the north. The smallest amount of data is at Tartu (TAR) (N = 232), and the highest is at Tromsø (TRO) (N = 68884). Stations in Svalbard were not included in these figures to make the polar plots easier to read. However, data from the Svalbard stations is shown in Fig. 14.

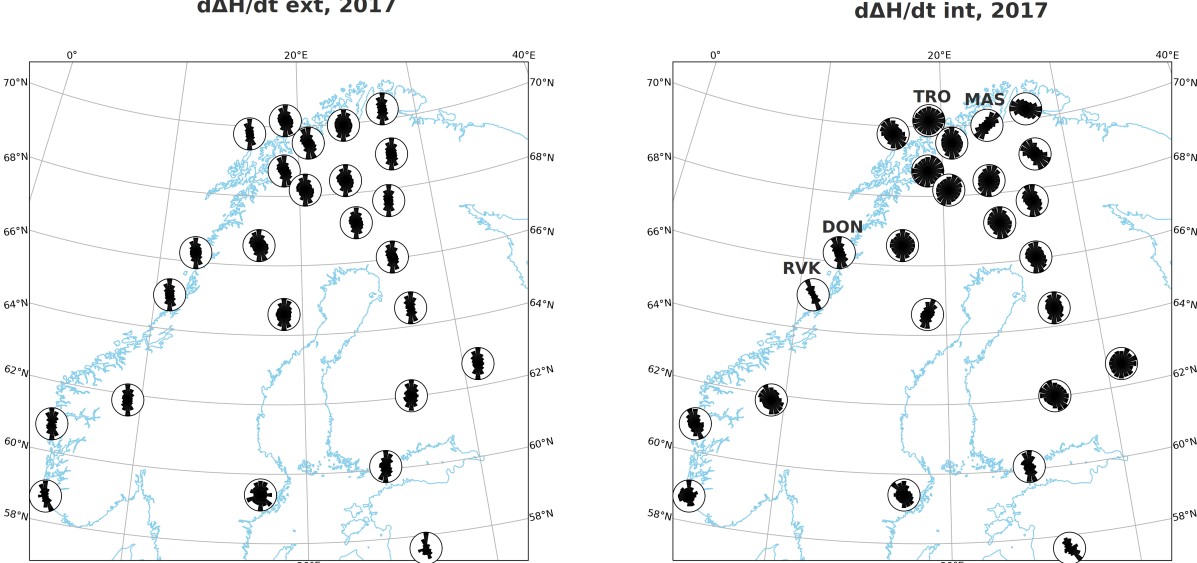

**Figure 5.** Directional distribution of external (left) and internal (right) $\mathrm{d}\Delta\mathbf{H}/\mathrm{d}t$ at IMAGE stations in 2017 when $|\mathrm{d}\Delta\mathbf{H}/\mathrm{d}t| > 1\ \mathrm{nTs^{-1}}$.

| KEV | MAS | TRO | AND | KIL | IVA | ABK | MUO | KIR | SOD |
|-----|-----|-----|-----|-----|-----|-----|-----|-----|-----|
| 45868 | 59583 | 68963 | 53507 | 52846 | 47798 | 49314 | 22370 | 34314 | 32443 |
| **PEL** | **JCK** | **DON** | **RAN** | **RVK** | **LYC** | **OUJ** | **MEK** | **HAN** | **DOB** |
| 31561 | 26815 | 32876 | 11942 | 17177 | 14902 | 7624 | 2231 | 2182 | 2750 |
| **SOL** | **NUR** | **UPS** | **KAR** | **TAR** | | | | | |
| 1676 | 1259 | 586 | 425 | 232 | | | | | |

**Table 2.** Number of 10-s data points in 2017 fitting the criterion $|\mathrm{d}\Delta\mathbf{H}/\mathrm{d}t| > 1\ \mathrm{nTs^{-1}}$ at each station, ordered by latitude, shown on the map in Figs. 4 and 5.

## 3.3 Yearly differences

The directional distributions of $\Delta\mathbf{H}$ were also analyzed yearly, to see if the solar cycle affects these distributions, or if certain years stand out. Number of stations used in the SECS field separation each year is shown in Fig. 6. The yearly polar plots for external and internal $\Delta\mathbf{H}$ for Sodankylä (SOD) are shown in Fig. 7. Same plots for the time derivative are shown in Fig. 8. Kevo station (KEV) shows some unexpected features that are shown in Fig. A1 in Appendix A.

The external and internal $\Delta\mathbf{H}$ do not show significant variation over the years. In the plots for the external $\Delta\mathbf{H}$ (Fig. 7 (a)) the clear southward orientation is visible each year. External and internal $\Delta\mathbf{H}$ also show some variation in south-east and

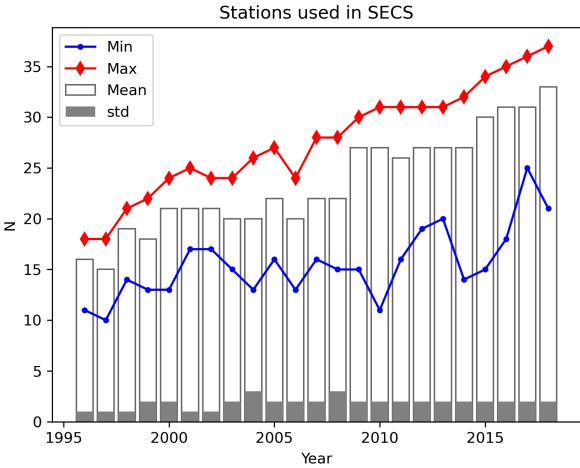

**Figure 6.** Mean, standard deviation, minimum and maximum number of stations used in SECS analysis per year. Numbers are calculated from daily values.

south-west directions. 1997 and 2004 seem to have equal amounts of southward and south-south-east oriented cases in external
$\Delta \mathbf{H}$. As for the internal $\Delta \mathbf{H}$, the years 1997 and 2004 do not stand out compared to the other years.

Plots of the external $\mathrm{d}\Delta \mathbf{H}/\mathrm{d}t$ (Fig. 8 (a)) do not show any clear differences between the years. The orientations are almost strictly northward-southward. There is a bit more variation to the east and west direction in 2012 and 2013. The polar plots for the time derivative of the internal $\Delta \mathbf{H}$ (Fig. 8 (b)) seems to be a bit more evenly distributed during the solar maximum years (2001, 2002 and 2012, 2013). The solar minimum years have more narrow distributions, especially 2007 and 2008.

Figure 9 shows the diurnal distribution of points fitting the criterion for $|\mathrm{d}\Delta \mathbf{H}/\mathrm{d}t|$ for SOD, 1996-2018. The time is expressed in magnetic local time (MLT), and each year is shown in a separate histogram. The histograms show that every year most events take place around the magnetic midnight or early morning hours. There is a clear minimum around noon/afternoon.

### 3.4 Mean directions

Figure 10 (left panel) shows the standard deviation and mean for $\Delta \mathbf{H}$ directions for each year at KIL, SOD and OUJ stations
for the external part (blue triangles) and internal part of (red dots). The error bars shows the standard deviation of the vector directions. The grey shading (OUJ, 2009) indicates very small amount of data, less than 100 10 s-data-points, fitting the derivative criterion that year.

No clear yearly trend is visible. The mean directions are strictly southward at KIL, SOD and OUJ for both external and internal parts of $\Delta \mathbf{H}$. Figure 10 (right panel) shows the mean directions for the external (blue triangles) $\mathrm{d}\Delta \mathbf{H}_{ext}/\mathrm{d}t$ and internal
(red dots) $\mathrm{d}\Delta \mathbf{H}_{int}/\mathrm{d}t$. There is only little variation in the mean directions. The solar minimum year, 2009, does stand out a bit, which may be due to a small number of large $|\mathrm{d}\Delta \mathbf{H}/\mathrm{d}t|$. The standard deviation range for KIL is 18-27°for $\Delta \mathbf{H}_{\mathbf{ext}}$, 30-

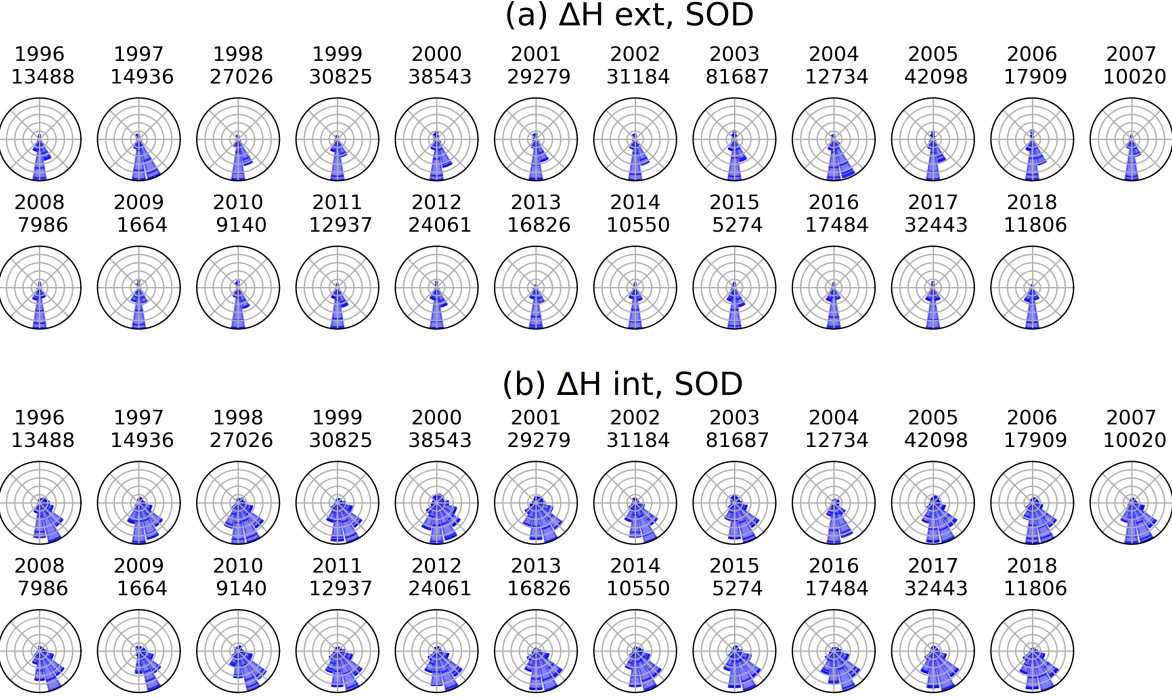

**Figure 7.** Directional distribution of (a) $\Delta\mathbf{H}_{ext}$ and (b) $\Delta\mathbf{H}_{int}$ at Sodankylä (SOD) between 1996-2018 when $|d\Delta\mathbf{H}/dt| > 1$ nTs$^{-1}$. The number of data points is plotted below the year label.

37°for $\Delta\mathbf{H}_{int}$, 40-50°for d$\Delta\mathbf{H}_{ext}$/d$t$ and 45 - 54°d$\Delta\mathbf{H}_{int}$/d$t$. For OUJ the ranges are: 15-23°for $\Delta\mathbf{H}_{ext}$, 24-37°for $\Delta\mathbf{H}_{int}$, 34-48°for d$\Delta\mathbf{H}_{ext}$/d$t$ and 39-52°for d$\Delta\mathbf{H}_{int}$/d$t$. For SOD the ranges are: 17-25°for $\Delta\mathbf{H}_{\mathbf{ext}}$, 27-37°for $\Delta\mathbf{H}_{\mathbf{int}}$, 36-48°for d$\Delta\mathbf{H}_{ext}$/d$t$ and 41-53°d$\Delta\mathbf{H}_{int}$/d$t$. There is no yearly trend visible, but the standard deviations at every studied station are the highest with d$\Delta\mathbf{H}_{int}$/d$t$.

### 3.5 Effect of T

We also studied how the time, $T$, over which the change in $\Delta\mathbf{H}$ direction is considered, affects the standard deviations of $\Delta\theta$. The goal was to figure out whether it is possible to find a characteristic time scale for the magnetic field. In other words: does the standard deviation of $\Delta\theta$ of the magnetic field (or the time derivative) reach an asymptotic value as $T$ increases? And if so, what is a typical time scale?

Figures 11 and 12 show the standard deviation of $\Delta\theta$ for the horizontal magnetic field and its time derivative respectively. There is a clear difference in their behavior. The standard deviation of $\Delta\theta$ of $\Delta\mathbf{H}$ is increasing faster when $T < 30$ min. After that, the increase is less steep, but there is no asymptotic value reached even after several hours. This behavior is similar with both the external and internal $\Delta\mathbf{H}$, although, with increasing values of $T$ the difference between the external and internal field becomes larger.

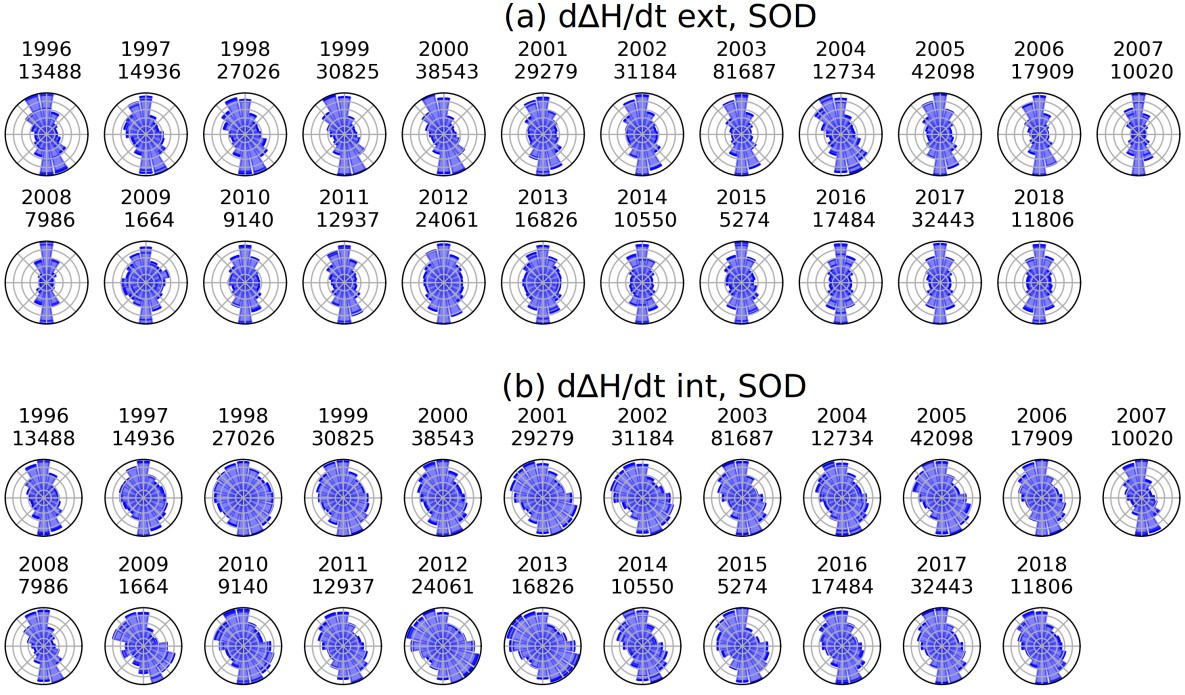

**Figure 8.** Directional distribution of (a) $d\mathbf{\Delta H}_{ext}/dt$ and (b) $d\mathbf{\Delta H}_{int}/dt$ at Sodankylä (SOD) 1996–2018 when $|d\mathbf{\Delta H}/dt| > 1\,\mathrm{nTs}^{-1}$. The number of data points is plotted below the year label.

For $d\mathbf{\Delta H}/dt$, an asymptotic value is reached quickly, just after about two minutes. This is seen with both the external and internal $d\mathbf{\Delta H}/dt$, but the difference between them is larger at small values of $T$, where the internal $d\mathbf{\Delta H}/dt$ tends to have slightly larger standard deviations. This behavior was seen at all the studied stations.

Also considering the mean value, mean($|\Delta\theta|$), instead of std($\Delta\theta$), yields similar results: An asymptotic value with $d\mathbf{\Delta H}/dt$ is reached around $T = 2$ min. With mean($|\Delta\theta(d\mathbf{\Delta H}/dt)|$) this asymptotic value is around 90 degrees, which is the mean of an even distribution in $\Delta\theta$. For the case of mean($|\Delta\theta(\mathbf{\Delta H})|$) there is no asymptotic value reached. These results are not shown in this paper.

Examples of distribution histograms at Kiruna (KIR), for different values of $T$, are presented in Fig. 13. The figure shows the distributions of $\Delta\theta$ for the external $\mathbf{\Delta H}$ (left panel) and its time derivative (right panel). Starting from the top panel, we have used $T = 10$ s, $T = 30$ s, $T = 10$ min and $T = 5$ h. In the plots for the external $\mathbf{\Delta H}$ the distributions slowly even out at larger values of $T$. Also, we see that in the lowest panel ($T = 5$ h) large values (+/- 180°) of $\Delta\theta$ become increasingly common. This means that the field is often pointing to the opposite direction after 5 hours. In the plots for the external $d\mathbf{\Delta H}/dt$, the distributions even out very quickly at larger $T$ values. Already at $T = 30$ s the distribution looks quite even.

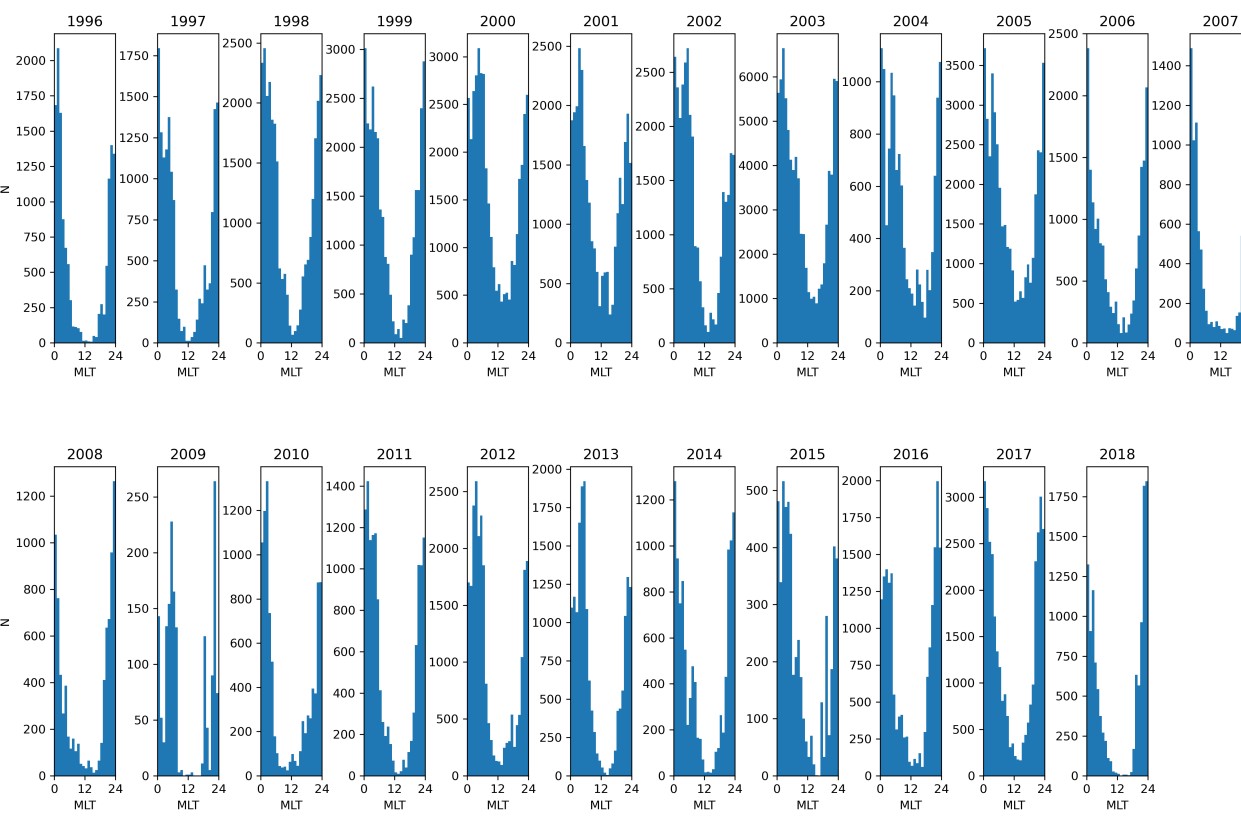

**Figure 9.** MLT distribution of the number of events (N) fitting the $|\mathrm{d}\Delta\mathbf{H}/\mathrm{d}t| > 1\ \mathrm{nTs}^{-1}$ criterion. Sodankylä (SOD), 1996–2018.

Figure 14 demonstrates values of the standard deviation of $\Delta\theta$ for external $\mathrm{d}\Delta\mathbf{H}/\mathrm{d}t$ at magnetometer stations, when $T =$ 10 min. The values are similar at all stations ranging from 104 to 110 degrees. They all are close to the theoretical standard deviation of an even distribution, which is described in detail in the Discussion section.

Finally we look at how the field strength changes over a period $T$. This is done by taking the ratio between the field amplitude at $t_0 + T$ and $t_0$, $t_0$ being the time when $\mathrm{d}\Delta\mathbf{H}/\mathrm{d}t$ reaches the threshold value ($1\ \mathrm{nTs}^{-1}$). These results are shown in Fig. 15. The ratios are below 100%, meaning that the derivative field typically decreases in amplitude after reaching the limit value ($1\ \mathrm{nTs}^{-1}$). The standard deviation is the smallest at the shortest time period, $T = 10$ s.

### 3.6 Effect of $\mathrm{d}\Delta\mathbf{H}/\mathrm{d}t$ activity level

Effect of a smaller threshold value for the time derivative was also studied. The other threshold that we used is $0.5\ \mathrm{nTs}^{-1} < |\mathrm{d}\Delta\mathbf{H}/\mathrm{d}t| < 1\ \mathrm{nTs}^{-1}$. Figure B2 in Appendix B shows the standard deviations of $\Delta\theta$ at different values of $T$, using smaller threshold. Overall, we get very similar results for these less active cases (i.e. similar asymptotic value) in the study of $\Delta\theta$.

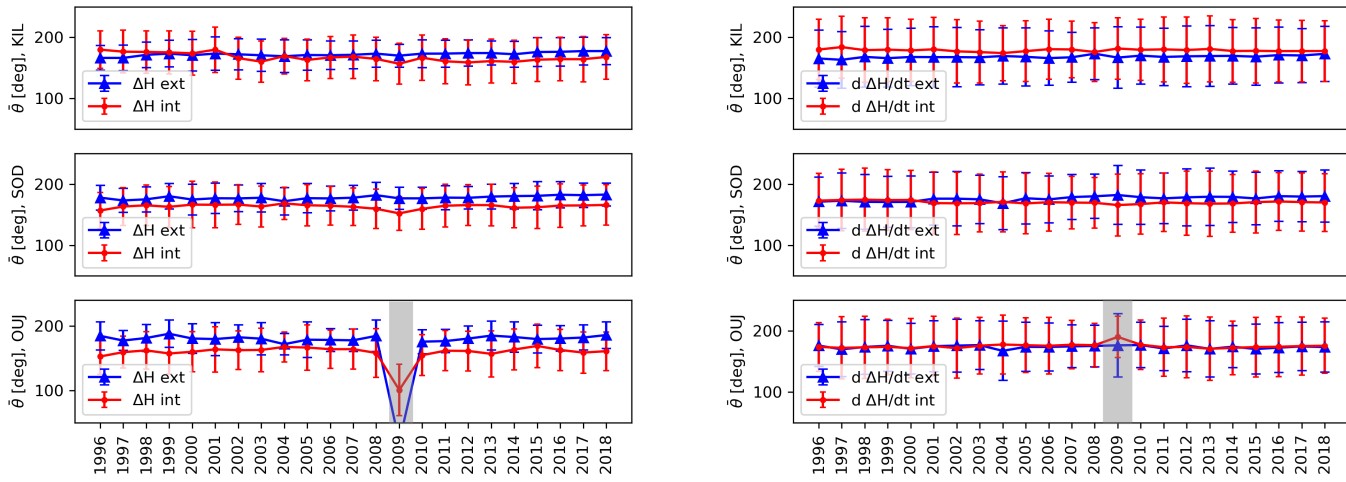

**Figure 10.** Mean directions, $\bar{\theta}$, and standard deviations (error bars) of external and internal $\Delta\mathbf{H}$ (left panel) and $\mathrm{d}\Delta\mathbf{H}/\mathrm{d}t$ (right panel) as a function of year at KIL, SOD and OUJ, 1996-2018. $\Delta\mathbf{H}_{ext}$ is marked with blue triangles and $\Delta\mathbf{H}_{int}$ with red dots. The grey shading indicates very few events (less than 100) fitting the criterion in OUJ, 2009.

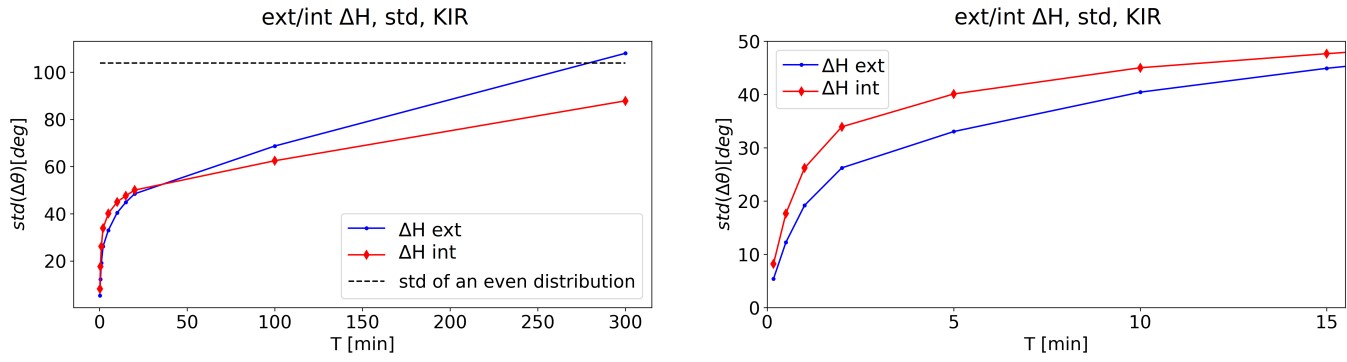

**Figure 11.** Standard deviations of $\Delta\theta$ for the external (blue line with dot markers) and internal (red line with diamond markers) $\Delta\mathbf{H}$ as a function of $T$ at Kiruna (KIR). Threshold value for chosen events is $|\mathrm{d}\Delta\mathbf{H}/\mathrm{d}t| > 1\ \mathrm{nTs}^{-1}$. On the left, $T$ range is from 0 to 300 min, a closeup on the first 15 min is shown on the right.

## 4 Discussion

### 4.1 Magnetic field separation

In this analysis we studied the directional distributions and change in the direction of the separated horizontal magnetic field and its time derivative. The separation was done to better understand the dynamics behind large GIC events. Previous studies

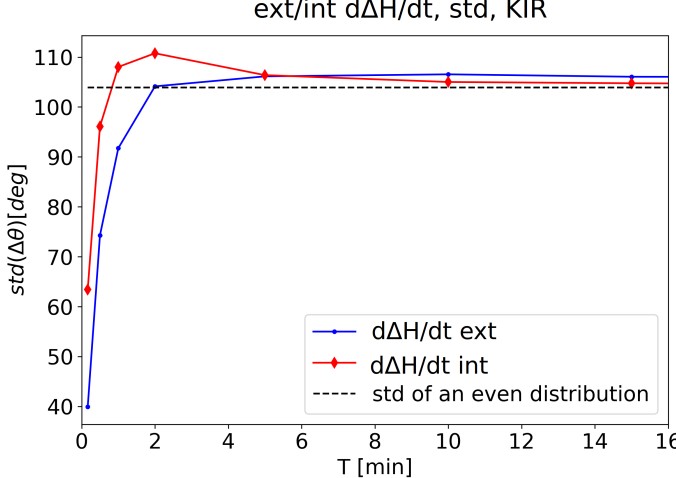

**Figure 12.** Standard deviations of $\Delta\theta$ for the external (blue line with dot markers) and internal (red line with diamond markers) d$\Delta\mathbf{H}$/d$t$ as a function of $T$ at Kiruna (KIR). Threshold value for chosen events is $|\text{d}\Delta\mathbf{H}/\text{d}t| > 1\ \text{nTs}^{-1}$.

have shown that d$\Delta\mathbf{B}$/d$t$ is a good indicator for GIC. Separating the field makes it possible to study individual contributions
of the external and internal fields.

The separation of the geomagnetic field can be done using several different methods, and each of them has their own advantages and disadvantages (e.g., Torta, 2020). The separation of the fields is never fully accurate, and there will be a small portion of the true external field present in the modelled internal field, and vice versa. The effect of using the 2D SECS method for the separation should be considered. It is possible that some of the effects seen in this analysis, could be produced by the
method. This could be verified in future studies, by repeating this analysis using a different method for the field separation. Also, the number and density of magnetometer stations has changed over the studied period, which may also affect the accuracy of the field separation, as dicussed by Juusola et al. (2020, Section 4.3). Implementing another separation method does not affect these sources of error.

However, the internal part of the separated field has been shown to follow the well known structure of the ground conductivity
(Juusola et al., 2020). For example, in Fig. 5 (right panel, internal field) the coastal effect is clearly visible at stations in the Norwegian coastline. Also, correlation between the electrojet currents derived simultaneously from IMAGE and low-orbit satellite have been shown to significantly improve when the separation is carried out (Juusola et al., 2016). Also, since the number of available stations has increased significantly over the years (see Fig. 6 for reference), but there is no visible difference in, e.g. the yearly mean directions (Fig. 10), this suggests that the number of stations used in SECS separation does
not significantly affect our results. Also in Juusola et al. (2020, Section 4.3) the authors performed a simple analysis on the reliability of the SECS separation by decreasing the density of stations used in the analysis. Their main conclusion was that even though there is a small increase in the internal contribution with the reduced network, the relative behavior of the different parameters is unchanged. These facts indicate that the separation should be fairly reliable.

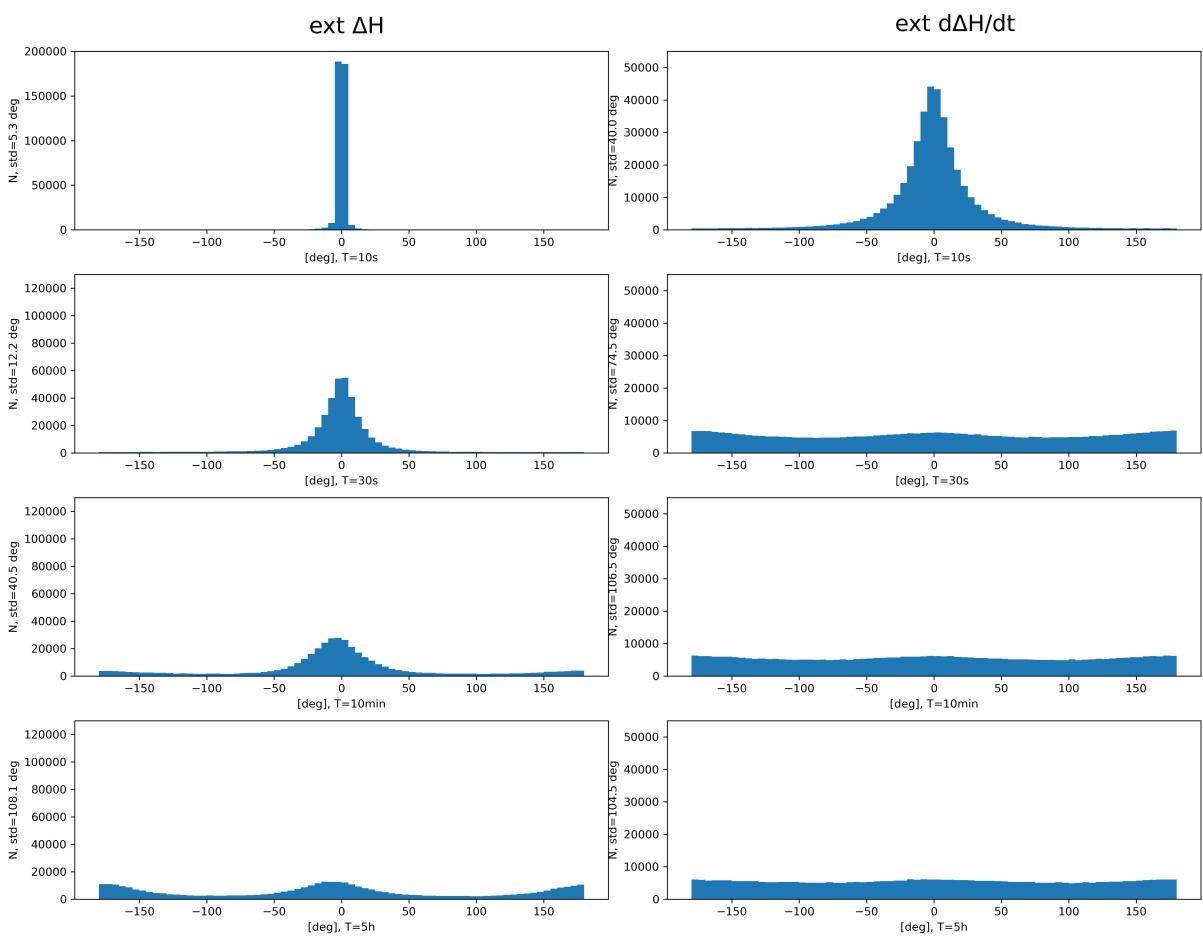

**Figure 13.** Examples of $\Delta\theta$ distributions of external $\Delta\mathbf{H}$ (left panel) and external $d\Delta\mathbf{H}/dt$ (right panel) at Kiruna (KIR) with different values of $T$. From top to bottom: $T$ = 10 sec, 30 sec, 10 min and 5 hours. The distributions even out at greater $T$ values.

## 4.2 Directional distributions

The majority of the events chosen with the derivative criterion have a clear southward distribution of $\Delta\mathbf{H}$, as seen in Figs. 4 and 10, which is produced by the westward electrojet. Effect of the eastward electrojet (northward distributions) is only visible at the southernmost stations. Also the directional distributions of $d\Delta\mathbf{H}/dt$ (Fig. 5) show the north-south orientation, although more scattered. This is not a new result, and has been described in previous studies. For example, Viljanen et al. (2001) had very similar results regarding the directional distribution of $d\Delta\mathbf{H}/dt$: mainly southward $\Delta\mathbf{H}$, with $|d\mathbf{H}/dt| > 1$ nTs$^{-1}$, and

a lot more scattered directional distributions for the time derivative. However, Viljanen et al. (2001) considered the total field ($d\Delta\mathbf{H}/dt$), so they could not discuss the ionospheric and telluric contributions separately.

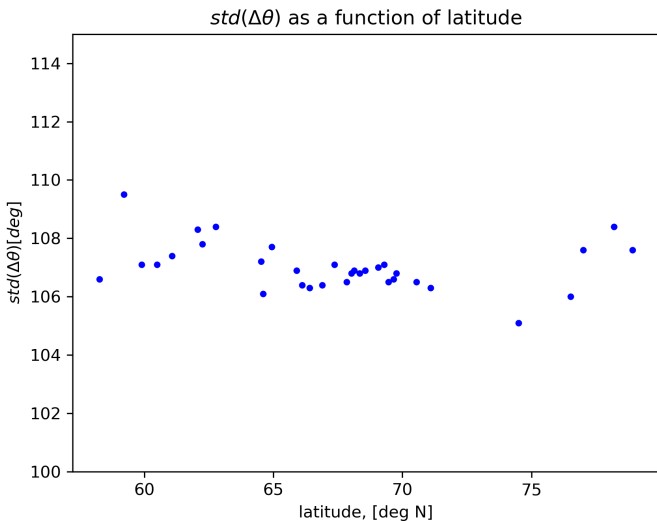

**Figure 14.** Standard deviation of $\Delta\theta$ for external $d\Delta\mathbf{H}/dt$ at IMAGE stations as a function of latitude. Data from 1996 to 2018, and $T = 10$ min.

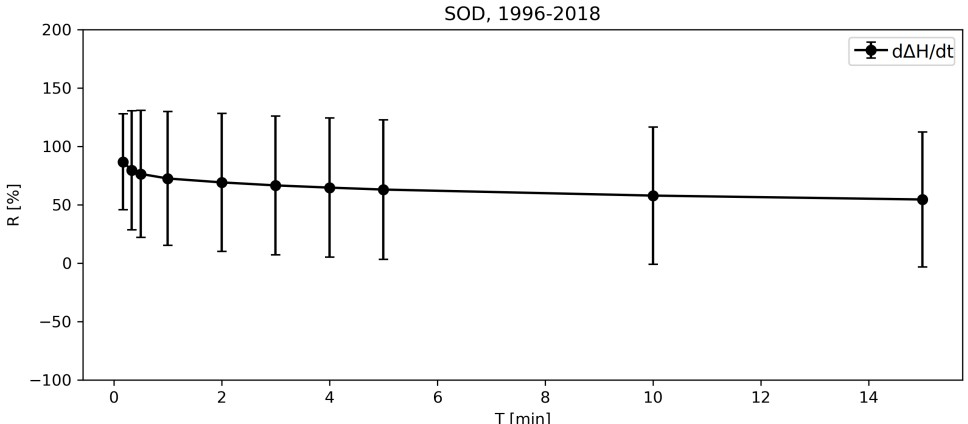

**Figure 15.** Mean values (black markers) and standard deviation (bars) of relative change in amplitude, R($T$), of total $d\Delta\mathbf{H}/dt$ at SOD. Data from 1996 to 2018, and $T = 10$ s ... 15 min.

We also noticed clear differences between magnetometer stations located at similar latitudes with $d\Delta\mathbf{H}_{int}/dt$ (Fig 5, right panel). The station specific differences with directional distributions near the Norwegian coastline (e.g. DON, RVK) are likely due to the local conductivity differences caused by the highly conducting seawater, also known as *the coast effect* (Lilley, 2007). These stations have a directional distribution with a component perpendicular to the coastline. The fact that this phenomenon is visible in the separated internal magnetic field, also shows the reliability of the used 2D SECS method. However, e.g. Masi

(MAS), which is located inland, also has a narrow distribution, which is known to be due to highly conducting, near-surface structures that strongly affect the geomagnetic field (Viljanen et al., 1995).

## 4.3 Effect of T

One of the main, new discoveries in this research, was the asymptotic value and characteristic time scale of the derivative vector. The asymptotic values of the standard deviations of $\Delta\theta$ for external and internal $\mathrm{d}\Delta\mathbf{H}/\mathrm{d}t$, can be explained via the value distributions and theoretical value for a uniform distribution.

Standard deviation, $\sigma$, for the uniform distribution between values $a$ and $b$, is given by equation 1. This is easily proven with basic equations for variance and probability density of a uniform distribution (Bertsekas and Tsitsiklis, 2008). In our study, where the magnetic field direction values range from a = -180 to b = 180 degrees, this theoretical value is approximately 104°:

$$\sigma = \frac{b-a}{\sqrt{12}} = \frac{360}{\sqrt{12}} \approx 103.9 \tag{1}$$

This value is close to the asymptotic values we got for the standard deviation of $\Delta\theta$ of $\mathrm{d}\Delta\mathbf{H}/\mathrm{d}t$, ranging from 104 to 110 for the studied stations. Values significantly above 104°, as is the case for large $T$ values for $std(\Delta\theta)$ of $\Delta\mathbf{H}$, indicate that the distribution is not uniform. This is evident in Fig. 13, where the $\Delta\theta$ distribution of $\Delta\mathbf{H}$ at $T = 5$ h shows two peaks, one around 0° and another around +/- 180°. However, when using longer periods of $T$, we end up comparing entirely different events affected by different ionospheric current systems. This raises the question if it even makes sense to use such long periods for $T$.

Our analysis and that of Pulkkinen et al. (2006) both yield, through different methods, the similar two minute time scale for the behavior of $\mathrm{d}\Delta\mathbf{H}/\mathrm{d}t$. After this time, the behavior of $\mathrm{d}\Delta\mathbf{H}/\mathrm{d}t$ resembles that of white noise, i.e., any memory of the past is lost. It is not clear, though, why the critical time scale has this particular value. As stated by Pulkkinen et al. (2006), the scales are linked to the corresponding scales in the dynamics of the ionosphere-magnetosphere system, but the link is all but self-evident. The size, motion, and lifetime of the $\mathrm{d}\Delta\mathbf{H}/\mathrm{d}t$ structures may contribute to the observed time scale. Because of the highly variable ground conductivity, development of the external $\mathrm{d}\Delta\mathbf{H}/\mathrm{d}t$ structures is generally much smoother than that of the internal $\mathrm{d}\Delta\mathbf{H}/\mathrm{d}t$ structures (Juusola et al., 2020). This can also be seen in Fig. 12, where the standard deviation of $\Delta\theta$ for the internal $\mathrm{d}\Delta\mathbf{H}/\mathrm{d}t$ is clearly higher than that for the external $\mathrm{d}\Delta\mathbf{H}/\mathrm{d}t$ during the first few minutes. Also, the results of Weygand et al. (2021) may give some explanations for the time scale origins. They show that several types of phenomena associated with the westward electrojet and/or Harang current system may be responsible for sudden magnetic perturbations.

Also Belakhovsky et al. (2018) studied the directional variation of the horizontal magnetic field and its derivative. They used a so-called *RB parameter* to determine if the field is changing more in magnitude or in direction. This parameter is similar to the $\Delta\theta$ quantity used in our study. For example, in a 2D-case, $\mathbf{B}(t) = \{X, Y\}$ and length of time series $N$, the RB-parameter is given by (Du et al., 2005):

$$RB = 1 - \frac{1}{N}\sqrt{\left(\sum_{n=1}^{N}\cos_x\alpha\right)^2 + \left(\sum_{n=1}^{N}\cos_y\alpha\right)^2} \tag{2}$$

where the magnitude of magnetic disturbance is $|\Delta B| = \sqrt{\Delta X^2 + \Delta Y^2}$, and the directions $\cos_x\alpha = \Delta X/|\Delta B|$ and $\cos_y\alpha = \Delta Y/|\Delta B|$. They used the total variation field, and not the separated field like we do. Consistently with our study they dis-

covered that the directional variability of $d\Delta\mathbf{B}/dt$ is greater than that of the variation field, $\mathbf{B}$. This was explained by the small-scale currents structures, the non-stationary vortex structures created by the local field-aligned currents.

In addition to the change in direction, we also looked at how the amplitude of total $d\Delta\mathbf{H}/dt$ changes over the time period, $T$ (Fig. 15). The mean of the relative change in amplitude, R($T$), was below 100% at all studied values of $T$, meaning that the amplitude of the derivative tends to decrease soon (relative to the 10 s sample interval) after reaching the threshold value. This is reasonable since the derivative changes very rapidly, e.g. see the case study in Fig. 3 (4th panel), and it is rare for the derivative amplitude to remain at high values for long periods. This was also shown by Weygand et al. (2021). The standard deviation slightly increases when $T$ increases, meaning that variation in the amplitude is smallest immediately after the amplitude reaches the threshold value.

## 4.4 Effect of activity level

In the last part of our study we tested a smaller threshold value for the horizontal time derivative. This smaller limit seems to have no major impact on the main results, i.e. the characteristic time scale of the derivative vector, or the relative change in amplitude. Plots, using the smaller threshold value, for the standard deviation of $\Delta\theta$ are presented in Figs. B1 and B2, and R($T$) in Fig. B3 in Appendix B. This result implies that the characteristic time scale is not related only to the most active events, but is visible also during the less active periods. This means that there are inherent physical features in the solar wind, magnetospheric and ionospheric system dictating the time scale independently of the magnitude of ionospheric currents (or $|\mathbf{H}|$).

## 4.5 Forecasting H, dH/dt and GIC

As shown in previous studies, the temporal behavior of GIC typically follows $d\Delta\mathbf{B}/dt$ at a nearby location. So, $d\Delta\mathbf{B}/dt$ is a good proxy for GIC (Viljanen et al., 2001). Our results show, however, that $d\Delta\mathbf{H}/dt$ has a very short persistence in terms of its direction. A similar feature holds true for its magnitude, which changes very rapidly (Fig. 3), so persistence of large values is also short (cf. Weygand et al. (2021, Fig. 2)). As a simple forecast, we could say that if $\mathbf{H}$ is large now, then it will be large also after several minutes or even later, and its direction will not change much. On the other hand, given the present $d\mathbf{H}/dt$, we have practically no chance to anticipate its magnitude nor direction after a couple of minutes. These results agree with the statement by Pulkkinen et al. (2006): *$d\Delta\mathbf{B}/dt$ fluctuations are not even in principle predictable in a deterministic way; nature sets boundaries for the accuracy with which we can forecast the future.* Even though the temporal behavior of $d\Delta\mathbf{B}/dt$ may not be predictable, the probability of large amplitude fluctuations can still be assessed based on the overall geomagnetic activity. Large $|d\mathbf{H}/dt|$ values are generally related to large $|\mathbf{H}|$ as mentioned by Viljanen et al. (2001, p. 1110).

Forecast of $d\Delta\mathbf{H}/dt$ (and GIC) would require two things: prediction of the external $d\Delta\mathbf{H}/dt$ from observed solar wind driving of the Earth's magnetosphere and ionosphere and prediction of the induction in the conducting ground as driven by the dynamics of the ionospheric and magnetospheric current systems. The latter part is relatively well understood (e.g., Ivannikova et al. (2018); Marshalko et al. (2021)) and mainly hampered by insufficiently detailed models of the Earth's conductivity. The first part is still a challenge, but hopefully global simulations will at some point be able to provide this. The development of the

external $d\Delta\mathbf{H}/dt$ in the immediate future could maybe be predicted by observing the dynamics of the $d\Delta\mathbf{H}/dt$ structures, e.g. Apatenkov et al. (2020). Recent studies have demonstrated that nighttime magnetic perturbations at high latitudes can occur in association with a range of ionospheric current systems, geomagnetic conditions, and auroral structures, and can cover large, moving regions (diameters of hundreds of $\mathrm{km}$)(Ngwira et al., 2018; Engebretson et al., 2019a, b).

Concerning the ground magnetic field obtained from simulations, we can suggest a simple diagnostic test. Perform a similar analysis for the simulated $d\Delta\mathbf{H}/dt$, as we have done for the measured field. If the same behavior is found in the direction of $d\Delta\mathbf{H}/dt$, then the simulations obviously are on the right track in describing relevant physics correctly. As a side note, simulations provide primarily only the external part of the ground field. So, the separated external contribution, as applied in our study, is the proper reference from measurements. Also, it is worth noting that any small difference in timing or location in the simulations makes the comparison challenging.

Besides first principle simulations, empirical methods are also popular, but they too face problems with $d\Delta\mathbf{H}/dt$. As a single example, the lower auroral electrojet index (AL), related to the north component of H, can be reasonably well predicted as a time-series based on solar wind observations (Amariutei and Ganushkina, 2012). However, there is no corresponding success with $d\Delta\mathbf{H}/dt$ as shown, for example, by Wintoft et al. (2015). Instead trying to predict a time series, they considered the 30 min maximum of $|d\mathbf{H}/dt|$. This gives hint of expected GIC levels, although it cannot provide full estimation of the geoelectric field, since information of the direction of $d\Delta\mathbf{H}/dt$ is not given.

## 5 Conclusions

In this study we first looked at directional distributions of $\Delta\mathbf{H}$ and $d\Delta\mathbf{H}/dt$ separately for the external and internal magnetic fields. We discovered:

1. Mainly southward orientations with both $\Delta\mathbf{H}_{ext}$ and $\Delta\mathbf{H}_{int}$, related to the westward ionospheric currents. North-south orientations with $d\Delta\mathbf{H}_{ext}/dt$, and more scattered orientations of $d\Delta\mathbf{H}_{int}/dt$. This backs up and extends results from previous research (e.g. Viljanen et al. (2001); Viljanen and Tanskanen (2011); Juusola et al. (2020)).

2. Clear, station specific differences in the directional distribution of $d\Delta\mathbf{H}_{int}/dt$. These may be due to ground conductivity differences at the respective stations. Also the coastal effect, due to a large ground conductivity gradient across the coastline, is visible in the results tending to rotate $\Delta\mathbf{H}_{int}$ perpendicular to the coastline.

3. There is little variation in the directional distributions and mean directions between years. However, $d\Delta\mathbf{H}_{int}/dt$ has more scattered distributions than $d\Delta\mathbf{H}_{ext}/dt$.

In the last part of our analysis we studied the directional change of $\Delta\mathbf{H}$ over varying time periods, $T$, $\Delta\theta$, and its standard deviation. The main new result discovered in this analysis is the asymptotic value, of about 104-110°, for $\Delta\theta(d\Delta\mathbf{H}/dt)$ standard deviation. This was reached at about $T = 2$ min, and holds true for the external, internal and total $d\Delta\mathbf{H}/dt$. We understand this so that the direction of $d\Delta\mathbf{H}/dt$ is not predictable based on the previous values. In other words, the time derivative of the geomagnetic field quickly "loses" its memory.

*Code and data availability.* IMAGE data used in this study is available at the website: https://space.fmi.fi/image (IMAGE, 2021). The code
used to calculate magnetic local times is available at https://apexpy.readthedocs.io/en/latest/ (van der Meeren and Burrell, 2015)

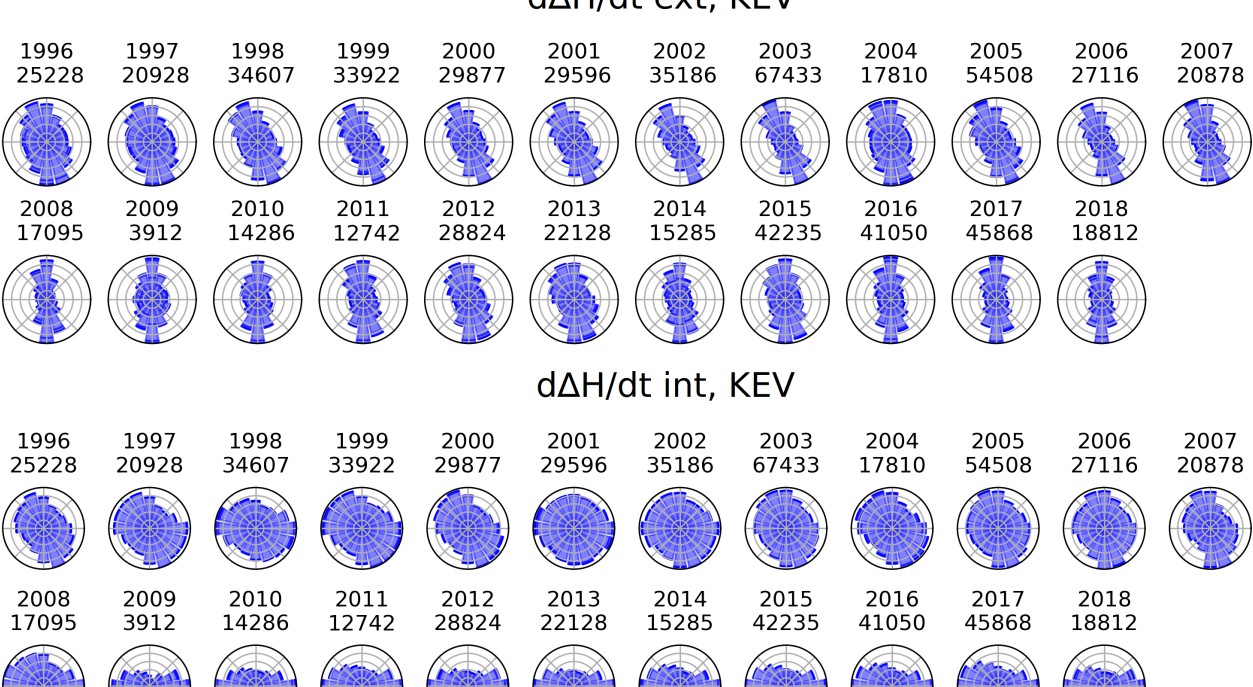

**Figure A1.** Yearly directional distributions of d$\Delta$**H**/d$t$ at Kevo (KEV), 1996-2008. Upper panel: external d$\Delta$**H**/d$t$, lower panel: internal d$\Delta$**H**/d$t$. The number of data points is plotted under the year label.

## Appendix A:  Yearly distributions, Kevo (KEV)

During this analysis we also discovered a curious feature in Kevo (KEV) internal d$\Delta$**H**/d$t$ directional distribution, presented in Fig. A1. The distribution of the internal d$\Delta$**H**/d$t$ rotates towards the east-west orientation in 2009 and stays like that until 2018. 2009 is one of the solar minimum years. There are significantly fewer data points that years. Amount of data drops from around 20000 to about 4000. However, the east-west orientation is visible even during the next solar maximum. The investigation for the reason behind this is still under way. Our best guess is that the tilt in the distribution could have been caused by the installation of a new device, or e.g. power cables, on the KEV research station in the beginning of 2009. More specifically, this happened in January or February 2009, as can be seen in Fig. A2. For this monthly plot we used a smaller threshold (d$\Delta$**H**/d$t > 0.5$ nTs$^{-1}$) to get more data points.

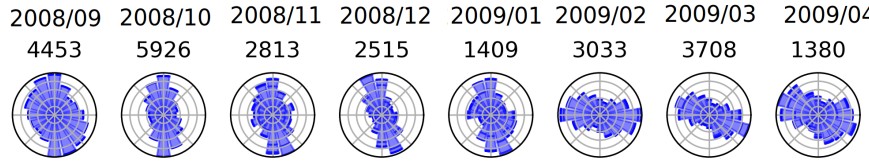

**Figure A2.** Monthly directional distributions of internal $\mathrm{d}\Delta\mathbf{H}/\mathrm{d}t$ at Kevo (KEV), 2008/09 to 2009/04. The number of data points is plotted under the year and month label. $|\mathrm{d}\Delta\mathbf{H}/\mathrm{d}t| > 0.5 \ \mathrm{nTs}^{-1}$.

**Appendix B: Effect of the activity level**

We also tested a smaller threshold value for the time derivative, $0.5 \ \mathrm{nTs}^{-1} < |\mathrm{d}\Delta\mathbf{H}/\mathrm{d}t| < 1 \ \mathrm{nTs}^{-1}$. In this section we repeat the analysis for the change in $\Delta\mathbf{H}$ and $\mathrm{d}\Delta\mathbf{H}/\mathrm{d}t$ direction, $\Delta\theta$ (Fig. B1), its standard deviation (Fig. B2) and relative change in amplitude, $R(T)$ (Fig. B3). There is no notable difference compared to the graphs made using the higher threshold.

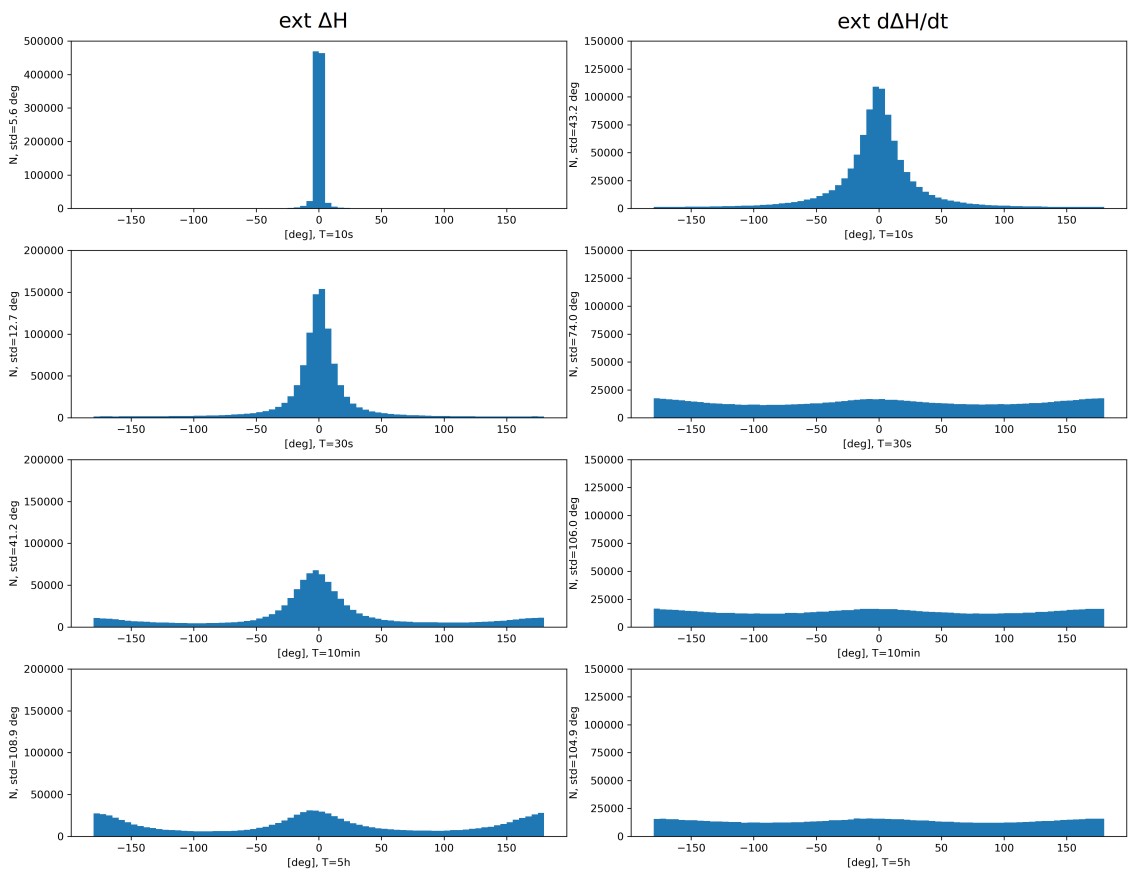

**Figure B1.** Histograms of $\Delta\theta$ at KIR at different time periods, $T$, using a threshold value $0.5 \text{ nTs}^{-1} < |\text{d}\Delta\mathbf{H}/\text{d}t| < 1 \text{ nTs}^{-1}$. On left: external $\Delta\mathbf{H}$, on right: external $\text{d}\Delta\mathbf{H}/\text{d}t$.

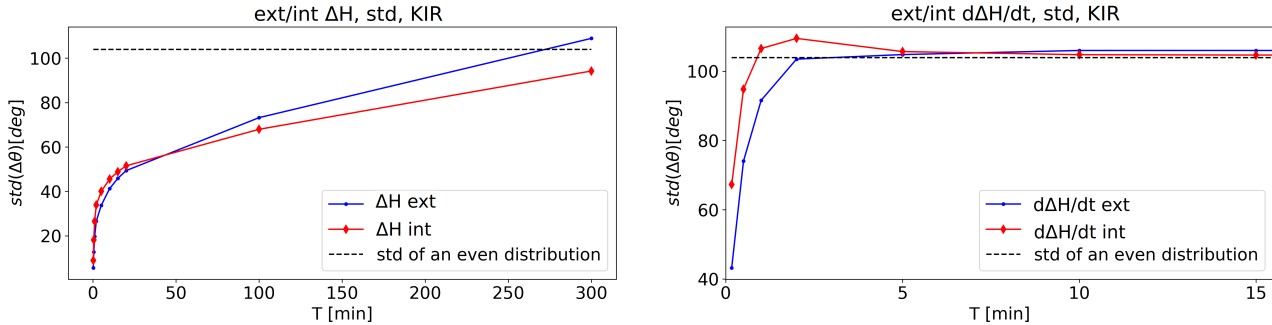

**Figure B2.** Standard deviation of $\Delta\theta$ at different time periods, $T$, using a threshold value $0.5 \text{ nTs}^{-1} < |\text{d}\Delta\mathbf{H}/\text{d}t| < 1 \text{ nTs}^{-1}$. Left panel: external and internal $\Delta\mathbf{H}$, right panel: external and internal $\text{d}\Delta\mathbf{H}/\text{d}t$.

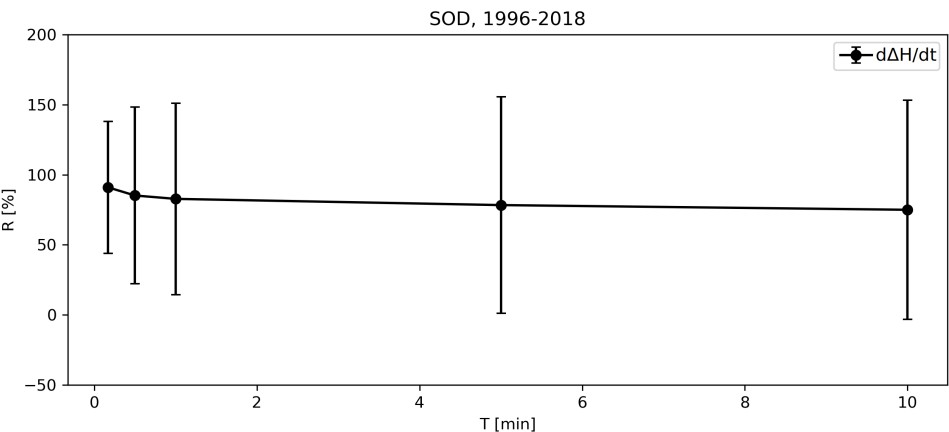

**Figure B3.** Mean values and standard deviation of R($T$) (relative change in amplitude) for total d$\Delta$**H**/d$t$ at SOD. Data from 1996 to 2018, and $T = 10$ s ... 10 min.

*Author contributions.* MK prepared most of the material and wrote the text with contributions from AV and LJ. SK provided help and ideas with data analysis and interpreting the results. AV and LJ provided expertise on the theoretical discussion.

*Competing interests.* The authors declare that they have no competing interests.

*Acknowledgements.* We thank Elena Marshalko for providing useful feedback for the article. We also thank Academy of Finland for funding this project (grant no. 314670 and 339329). Finally, we thank the institutes that maintain the IMAGE Magnetometer Array: Tromsø Geophysical Observatory of UiT the Arctic University of Norway (Norway), Finnish Meteorological Institute (Finland), Institute of Geophysics Polish Academy of Sciences (Poland), German Research Centre for Geosciences (GFZ, Germany), Geological Survey of Sweden (Sweden), Swedish Institute of Space Physics (Sweden), Sodankylä Geophysical Observatory of the University of Oulu (Finland), and Polar Geophysical Institute (Russia).

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
