# Peer review of "The time derivative of the geomagnetic field has a short memory"

_Annales Geophysicae, 2022_

## Author Response (AR1)

**Comment on angeo-2022-4, Referee Comment 1**

**We thank the referee for the useful comments regarding our manuscript and for stimulating further discussion based on relevant articles. Here are our specific responses and implemented revisions (in bold) based on referee comments:**

"The "main new result" of this paper, that the direction of the geomagnetic field time derivative has a very short "reset time," was anticipated by Belakhovsky et al. (2018), as the authors note,"

"… but also in significant detail in a recent study of similar events in Arctic Canada by Weygand et al. (2021) … "

> **These authors show examples illustrating how dB/dt varies clearly more rapidly in direction than B. However, no quantitative characteristic time scale is given in this paper.**
>
> **The time scale that Weygand et al. discuss is different from ours. However, their study provides useful insight which is relevant also to our manuscript.**
>
> ***Reference and/or discussion based on Weygand et al. (2021) is added in Results 3.1 L109 and Discussion 4.3 lines 239-241: "… Also, the results of Weygand et al. (2021) may give some explanations for the time scale origins. They show that several types of phenomena associated with the westward electrojet and/or Harang current system may be responsible for sudden magnetic perturbations.", also modified L255, L269.***

Specific Comments

> 1.Line 150: The text incorrectly states that "Figures 10 and 11 show $\Delta\theta$" but these figures and their captions make it clear that what is shown is the standard deviation of $\Delta\theta$, not $\Delta\theta$ .
>
> > **The figures show the standard deviation of $\Delta\theta$. The text was corrected accordingly, *L157: "Figures 10 and 11 show the standard deviation of $\Delta\theta$ …"***
>
> 2.Lines 152-153: Given the above confusion between $\Delta\theta$ and the standard deviation of $\Delta\theta$, it is not clear to this reviewer whether or not "standard deviation" belongs in this sentence. It is also not clear what is meant by their "mean values" yielding similar results. Over what variable and range are these mean values ($\Delta\theta$ or std ($\Delta\theta$)) calculated?
>
> > ***More thorough explanation added in Section 3.5, lines 165 – 167: "…considering the mean value, mean($|\Delta\theta|$), instead of std($\Delta\theta$), yields similar results: An asymptotic value with dH/dt is reached around T = 2 min. With***

*mean(|Δθ(dH/dt)|) this asymptotic value is around 90 degrees. For the case of mean(|Δθ(H)|) there is no asymptotic value reached. ..."*

3.Lines 185-186: The westward electrojet also produces southward magnetic field perturbations before magnetic midnight.  See, for example, Table 3 of the SECS analysis of large ( >6 nT/s) pre- and post-midnight magnetic field perturbations reported by Weygand et al. (2021).

**This is correct, and was also shown by Viljanen et al. (2001): Fig. 9.**
***Revised text: Section 4.2 lines 205-211. Mention of post-magnetic midnight was removed.***

4.Lines 215-218: The time scale of 80 s to 100 s for the behavior of dH/dt is clear in the Pulkkinen et al. (2006) paper, but is asserted without any specific documentation or quantification as being a result of the analysis presented in this paper.  This statement needs to either be adequately justified or removed.

***Section 4.3, lines 232 - was reformulated and more discussion added based on Weygand et al 2021. "Our analysis and that of Pulkkinen et al. (2006) both yield, through different methods, the similar two minute time scale for the behavior of dH/dt..."***

5.Lines 217-218: The manuscript does not provide any explanation for this time scale, other than that "The size, motion, and lifetime of the dH/dt structures may contribute to the observed time scale."  The Weygand et al. (2021) paper provides detailed information at higher time resolution than provided in this study that may be helpful in developing such an explanation.

**Additional discussion will be added based on Weygand et al (2021).**
***Section 4.3 lines 232-241, were reformulated and more discussion added based on Weygand et al 2021***

Figure 2 of Weygand et al. shows a histogram of the duration of all dB/dt derivative amplitudes above 6 nT/s observed at two Canadian stations during 2015.  The peak of the distribution of the durations of derivative amplitudes |dB/dt| ≥ 6 nT/s, which are different from the duration of the magnetic perturbations (ΔB), was between 10 and 15 s, but the range was between a few seconds (most common for MPEs with peaks only slightly above 6 nT/s) up to 71 s.

This figure, based on 10x higher sampling rate data than was used in this manuscript, provides a corrective to the statement in lines 232-233 that "the amplitude of the derivative tends to decrease immediately after reaching the threshold value."  The amplitude of course must increase immediately after reaching whatever threshold is used, whether 1 nT/s or 6 nT/s, if it is ever to reach a much higher value (which is often observed) but this figure quantifies the distribution of durations; it is short (not immediate) only relative to durations quantized by 10-s sampling.

***Clarification added in Section 4.3. L252-255: "... the amplitude of the derivative tends to decrease soon (relative to the 10 s sample interval) after reaching the threshold value. This is reasonable since the derivative changes very rapidly, e.g. see the case study in Fig. 3 (4th panel), and it is rare for the derivative amplitude to remain at high values for long periods. This was also shown by Weygand et al. (2021). ..."***

This rapid falloff of durations above 20 s provides a ready explanation (with a correction) for the statements in lines 230-233 and agrees with the statement on in lines 234-235 that it is rare for the derivative amplitude to remain at high values for long periods.

***Reference to Weygand et al 2021 was added Section 4.3 L255***

Weygand et al. (2021) also examined the dB/dt durations above 6 nT/s as a function of three categories of time delay Δtso after the most recent prior substorm.  For Δtso ≤ 30 min category the mean duration was 19.0 ± 0.9 s, for 30 < Δtso < 60 min the duration was 17.7 ± 2.1 s, and for Δtso ≥ 60 min the mean duration was 12.8 ± 1.8 s where the uncertainty given is the error of the mean.

In addition, Weygand et al. (2021) presented several example events, combining multistation magnetometer observations with SECS analyses and in some cases auroral images, that showed that short-lived and highly localized vertical currents and associated localized ionospheric currents were associated with large perturbations and dB/dt values at individual stations.

***Section 4.3. L232-257 were reformulated and additional discussion added based on Pulkkinen et al. (2006) and Weygand et al. 2021.***

The location of these currents relative to the measuring stations determined details of the orientation of the observed magnetic perturbations and their vector derivatives as well as the extent of their duration.  No issue of memory needs to be invoked.

**The "issue of memory" is merely a lighthearted and easily accessible way of describing one of our results. We find it important to emphasise the difference between H and dH/dt. H has a longer "memory", i.e. its direction changes clearly slowly than of dH/dt. As is visually obvious (as illustrated by Weygand et al.), the magnitude of H also changes slowly. So, if we know the present value of H, its (near-)future values (the next few minutes or later) will not be very different from the present. On the contrary, the next value of dH/dt (<1 min from the present) can be completely different, both by magnitude and direction.**

***Section 4.5 was re-written to include more discussion on simulations and the difficulty of predicting dH/dt and GIC based on Pulkkinen et al. (2006),Amariutei***

*and Ganushkina, 2012), Wintoft et al. (2015), Ngwira et al., 2018; Engebretson et al., 2019a, b)*

Technical Corrections

Line 209:  This line contains two minor errors.  First, as in line 150, the words "standard deviation of" need to be added before "Δθ."  Second, the values "104 to 110" do not agree with the values of "105 to 109" stated in line 155 in reference to Figure 13.

**Corrected both instances to read "104 to 110", L175, 227**

**Comment on angeo-2022-4, Referee Comment 2**

**We greatly appreciate the comments from the referee. They were extremely helpful in stimulating further discussion in the manuscript. Here are our specific responses and implemented revisions (in bold) based on referee comments:**

Even if this result is significant in itself, I have serious doubts that it is sufficiently relevant to merit an exclusive publication, especially if this subject has already been investigated by other researchers, who reached similar conclusions, though perhaps using different methods and parameters (Belakhovsky et al., 2018; Weygand et al., 2021, Pulkkinen et al., 2006).

**Belakhovsky et al. applied the RB parameter, which describes the variability of a vector in time and space. No characteristic time scales are determined through  this analysis. As previously replied to Ref-1, the time scale that Weygand et al. discuss is different from ours. They study the persistence of large time derivative values, whereas we study persistence in magnetic field directions. However, their study provides useful insight, which is relevant also to our manuscript. In the study by Pulkkinen et al., one difference in the methodology is that they consider the magnetometer network as a whole through a structure function, while we study explicitly H and dH/dt at single stations. Overall, they have a similar result using an entirely different method.**

Furthermore, the present study is based entirely on reliance on the external/internal separation of the geomagnetic field provided by the SECS technique; however, no assessment is made of the uncertainty of this separation, as the authors themselves acknowledge. For these reasons, I strongly recommend including additional, substantial material based on the following.

**Solar wind-magnetosphere-ionosphere interaction creates electric currents in the near-Earth space. The temporally varying "external" magnetic field of these currents drives induction in the conducting ground. The "internal" magnetic field of the induced electric currents is superposed to the external magnetic field, creating the measurable geomagnetic variations. If the driving external geomagnetic variations and a detailed model of the ground conductivity are given, 3D induction modeling can estimate the internal geomagnetic field and the geoelectric field (e.g., Marshalko et al., 2021). The geoelectric field is the driver of GIC in a conductor system, but the amplitude of the GIC is also affected by the parameters of the system.**

**Study of the characteristics of internal geomagnetic variations provides information on the complicated process of geomagnetic induction. Study of the characteristics of external geomagnetic variations, on the other hand, provides information on the solar wind-magnetosphere-ionosphere interaction. Global simulations typically estimate only this part of the geomagnetic field, and assessment of their abilities in this regard requires understanding of the characteristics of the external geomagnetic variations. Thus, study of the characteristics of both internal and external geomagnetic field can improve our ability to forecast space weather events that cause large GIC. The reliability of the method is discussed more in the specific comments below.**

**More discussion on the 2D SECS was added in section 4.1 lines 195-203:**

*"... Also, the number and density of magnetometer stations has changed over the studied period, which may also affect the accuracy of the field separation, as dicussed by Juusola et al. (2020). Implementing another separation method does not affect these sources of error.*

*However, the internal part of the separated field has been shown to follow the well known structure of the ground conductivity(Juusola et al., 2020). For example, in Fig. 5 (right panel, internal field) the coastal effect is clearly visible at stations in the*

*Norwegian coastline. Also, correlation between the electrojet currents derived simultaneously from IMAGE and low-orbit satellite have been shown to significantly improve when the separation is carried out (Juusola et al., 2016). These results indicate that the separation should be fairly reliable."*

**Specific comments:**

- The authors argue that their findings are important to the subject of GIC (this word is repeated a number of times along the manuscript) but they do not substantiate this argument based on GIC measurements of any type.

  **We thank the referee for stimulating a more detailed discussion on the relation between GIC and dB/dt, and of forecasting GIC (and dB/dt). As shown in many previous studies, the temporal behavior of GIC typically follows dB/dt at a nearby location. So, dB/dt is a good proxy. A full modeling requires determination of the geoelectric field and including a model of the conductor system in question, but this is beyond the scope of our manuscript.**

  **We have rewritten section 4.5 to include more discussion on the relation of dH/dt and GIC, and GIC forecasts based on Wintoft et al. (2015).**

  Could the authors provide evidence that their conclusions are somehow reflected in GIC measurements? For example, is it expected that the GIC (which depends on the time derivative of the horizontal magnetic field) has a typical lifetime of two minutes, comparable to the directional persistence of dH/dt?

  **When talking about a typical lifetime of GIC, some care is needed. An obvious choice is to consider the length of events when GIC exceeds a given threshold. The larger the threshold the longer the event. This was noted by Viljanen et al. (2014, Figs. 9-10) who considered the GIC sum in a large grid. Due to the close relation of GIC and dB/dt, we could as well consider the persistence of large dB/dt values. As expected, durations of large dB/dt events are short (e.g., Weygand et al., 2021, cf. comments by Ref-1). The same was also shown by Juusola et al. (2015) in terms of the time derivatives of equivalent currents.**

  **We have rewritten section 4.5 to include more discussion based on Weygand et al. 2021, and on GIC forecasts based on Wintoft et al. (2015).**

  If not, is it because the infrastructures (e.g., power network or oil/gas pipelines) where the GIC is expected to flow have not a preferent direction (e.g., N-S or E-W)?

**There seems to be no specific direction to which a conductor system is most sensitive to GIC, at least at higher latitudes such as the IMAGE magnetometer network. Since the directional distribution of the total dH/dt (ext+int) is very scattered then the same holds true for the geoelectric field.**

Also, the authors point that the final aim is to forecast GIC. I guess GIC can be predicted by trying to anticipate the ground magnetic variations based on IMF/solar wind observations, along with accurate models of the ground conductivity. Can they specify more clearly how is it expected that the main conclusion of the manuscript (i.e., the "short memory" of the time derivative) helps in this endeavor?

**Ironically, our result does not evidently help in deterministic forecasting! We can only agree with Pulkkinen et al. (2006) that dB/dt "fluctuations are not even in principle predictable in a deterministic way; nature sets boundaries for the accuracy with which we can forecast the future"**

**There is a possibly interesting spin-off concerning simulations. A simulated ground magnetic field and its time derivative should show similar features as the measured field. So we could repeat our analysis for simulated fields and especially check whether the same time scale for dB/dt appears. If not then some fundamental physics is missing in the simulation. As a side note, simulations provide primarily only the external part of the ground field. So a proper reference from measurements is the separated external contribution.**

**Section 4.5 was re-written to include more discussion on simulations and the difficulty of predicting dH/dt and GIC based on Pulkkinen et al. (2006),Amariutei and Ganushkina, 2012), Wintoft et al. (2015), Ngwira et al., 2018; Engebretson et al., 2019a, b)**

Perhaps they refer to "evaluating a potential GIC risk level by means of the dH/dt proxy" rather than "forecasting GIC"?

**Actually not. GIC risk level is obviously very much related to the magnitude of dH/dt (and the geoelectric field). This is a different aspect than trying to understand why dH/dt has a very complex behavior. Citing Pulkkinen et al. (2006, paragraph 42): "The most dramatic change in the observed dynamics occurred in the dBx/dt and dBy/dt fluctuations at temporal scales between 80 < t < 100 s. These scales are**

**naturally linked to corresponding scales in the dynamics of the ionosphere-magnetosphere system. However, the link is all but self-evident and we postpone further speculations to forthcoming investigations." It seems that "the link" is still quite much unsolved.**

**More discussion added in Section 4.3 lines 234-236: "It is not clear, though, why the critical time scale has this particular value. As stated by Pulkkinen et al. (2006), the scales are linked to the corresponding scales in the dynamics of the ionosphere-magnetosphere system, but the link is all but self-evident."**

- The present study is entirely based on the reliance on the external/internal separation of the geomagnetic field provided by the SECS technique;

    **Even if there are uncertainties in the field separation, the main result concerning the time scale of dH/dt does not change. It could also be determined without the field separation. E.g. Δθ timescale is visible in both, external and internal dH/dt, so it is also visible in the total dH/dt.**

    **More discussion on the 2D SECS was added in section 4.1 lines 195-203.**

    however, the effectiveness of this method is subject to different aspects, such as the nature of the primary/secondary sources, the density of ground magnetometers, or the election of the cutoff parameter for the singular values of the singular value decomposition (SVD) typically used in the context of SECS, among others. The authors have ample experience on this technique, so they should be able to provide an estimate of the uncertainty of the modeled magnetic field and of its external/internal separation in particular. I'm not aware of many articles where this important subject is treated, but perhaps the following thesis can help: https://open.uct.ac.za/handle/11427/35593 (see section 7.2). If this implementation is not feasible, I encourage the authors to at least apply the alternative method of field separation they refer to in Section 4.1 in order to assess how much of the separation depends on the method utilized.

    **In our implementation of the 2D SECS method, the cutoff parameter for singular values of the singular values decomposition is zero. As a consequence, all components of the observed geomagnetic field are perfectly reproduced at all stations used in the analysis. In this study we do not use interpolated values between stations, so considering the**

**reliability of the solution away from the stations, as is in done in the thesis suggested by the reviewer, would not help. Measurement errors of the magnetometers are also very small. Of course, the separation of the geomagnetic field is not perfect and is affected by the density of the magnetometers as well as the boundary conditions, as discussed by Juusola et al. (2020). Unfortunately, estimating this uncertainty is not at all simple. Implementing another separation method does not affect these sources of error.**

**The internal part of the separated field has been shown to follow well the known structure of the ground conductivity (Juusola et al., 2020) and correlation between the electrojet currents derived simultaneously from IMAGE and low-orbit satellite have been shown to significantly improve when the separation is carried out (Juusola et al., 2016). These results indicate that the separation should be fairly reliable.**

**We included more explanation and discussion on the reliability of the 2D SECS method in Sections 2.1 lines 62–66, 4.1 lines 195-198, 4.2 lines 212 - 216.**

**Technical corrections:**

- L19: I would suggest: "Space weather events, eventually produced by eruptive phenomena in the Sun, can have harmful effects on Earth, for example via ..."

  - **OK, L20 changed as suggested**

- L33: Faraday's induction law.

  - **OK, L33 changed as suggested**

- Figure 1 is somewhat naïve, and in my opinion unnecessary -just consider my comment as a recommendation. In its place (though perhaps not as Figure 1), I would find more useful to illustrate the concept of $\Delta\theta(H)$ and, if possible, that of $\Delta\theta(dH/dt)$, which is central to this manuscript. I think the horizontal projection of the geomagnetic field can be represented at times t and t + T as two arrows, and then represent the corresponding variation in $\theta$.

  - **Figure 1 (SECS method) was removed. The first figure is now of the IMAGE stations on map. Added Fig. 2 of $\Delta\theta(H)$, as suggested.**

- L62 "2D SECS": I guess you have used internal and external nodes for the field separation. Is there a specific designation for this modality to differentiate it

from the use of external nodes only (which would be the case to model the total horizontal field when there is no need for external/internal separation)?

- **We have added a longer description of the method in Section 2.1 L62-66: "... *In this method there are two layers of elementary currents used, one in the ionosphere (90 km altitude) and the other just below the ground (0 km, for numerical reasons set to 1 m). In our implementation of the 2D SECS method, the cutoff parameter for singular values of the singular values decomposition is zero. As a consequence, all components of the observed geomagnetic field are perfectly reproduced at all stations used in the analysis.*"**

- L75: The IMAGE time resolution is 10 s. Does the threshold of 1 nT/s refer to a mean variation computed as 10 nT in those 10 s? If so, I think the authors should state it.

  - **Clarification on L78-79: "... Since the used data is 10 s data, this limit value for the derivative means that the change in its amplitude is above 10 nT per 10 s."**

- The authors always refer to Bx, By and H whereas section 2.1 specifies that baselines are subtracted from the data using a certain automatic method. In consequence, they work with variations of those quantities. I think this point is important and the nomenclature currently used may give rise to confusion. Properly speaking, the studied quantities are ΔBx, ΔBy and ΔH (where, e.g., ΔH = H – Hb, with Hb the baseline value). I strongly recommend using the deltas before these quantities everywhere.

  - **Ignoring Δ is a common practice within space physics community, and makes notations a little simpler. Mention of this is added in L68-69: "We use a simple notation for the baseline subtracted data: H = H measured − H baseline ."**

- Table 1, in 2nd line replace H with |ΔH|; and in 4th line replace dH/dt with | dH/dt|.

  - **Table 1 was changed as suggested**

- L96: Bx and By should not be in bold face. Idem for caption of Figure 3.

  - **L103 and caption of Fig. 3 were corrected as suggested**

- L98: Replace dH/dt with |dH/dt|.

  - **OK, L104 changed as suggested**

- Caption of Figure 3: I would recommend to state "Figure 3. Plot of different quantities related with the horizontal magnetic field at Tromsø, ...". 4) Amplitude of the time derivative ...
  - **Caption of Fig. 3 changed as suggested**
- L116: No mention is made of stations in Svalbard and surroundings, which do not appear in Figures 4, 5 and Table 2 (and indeed anywhere except for the map in Figure 2). Are they only used for the purpose of the SECS-based external/internal separation?
  - **Mention of this was added on L126-127: "... *Plots for stations in Svalbard were not included in these figures to make the polar plots easier to read. However, data from the Svalbard stations is shown in Fig. 13.*"**
- Caption of Table 2. Say the stations are ordered by latitude.
  - **Caption of Table 2 changed as suggested**
- L124: "... over the years"?
  - **L131 changed as suggested**
- Figure 6: The number of data points in SOD for 2017 shows 32443 against the 32436 of Table 2. Isn't that an inconsistency?
  - **There was an inconsistency of one day in the used time range. This was corrected, and values in Table 2 updated accordingly.**
- Figure 7 and others: I would use "dH/dt" instead of "dH", as in the text.
  - **Titles in Figs. 5,7,12 and Appendix A1, B1 changed as suggested**
- Draw the line corresponding to the even distribution of Δθ in figure 10a.
  - **Figure 10 modified as suggested**
- L149: Figures 10 and 11 show the standard deviation of ...
  - **L157 changed as suggested**
- What is the meaning of the last sentence in the paragraph L149 – L153?
  - **Clarification added on L165-167: "*Also considering the mean value, mean(|Δθ|), instead of std(Δθ), yields similar results: An asymptotic value with dH/dt is reached around T = 2 min. With mean(|Δθ(dH/dt)|) this asymptotic value is around 90 degrees. For the case of mean(|Δθ(H)|) there is no asymptotic value reached.*"**

- L154: Figure 13 is referred before Figure 12. I would recommend following the logical ordering.

  - **OK, Text was revised L168-174**

- Figure 8: Show a title for the x- and y-axis for at least one of the subplots, e.g., "MLT (h)" and "# of events". Also, MLT = 25 sounds bad. Please, place ticks at 0, 12 and 24 h.

  - **Figure 8 modified as suggested**

- Figure 9: y-axis is missing a "mean θ" (or equivalent) followed by the station name, e.g., <θ> KIL.

  - **Labels in Fig. 9 modified**

- Figures 10, 11 and 13: Likewise, y-axis is missing a "Δθ".

  - **Labels in Figs. 10,11,13 modified**

- Figure 14: y-axis is missing an "R". Caption: Specify that the bars indicate the std of R.

  - **y-label in Fig. 14 modified**

- Paragraph L185-189: Only Figure 8 is mentioned, but the fact that the magnetic field is predominantly southward is shown in Figure 9.

  - **Text was modified to mention Fig. 9 L205-206**

- L191-193: Please, refer to a specific figure the reader should look at. Do the authors refer to Figure 5 (right) here? If so, I don't see an especially narrow distribution at MAS station (unless I get confused with nearby stations); instead, other nearby stations like IVA show a yet narrower distribution.

  - **L212 modified to specify Fig. 5 right panel, also the discussed stations are now marked on the map in Fig. 5 right panel.**

- In the context of the discussion of the coast effect (L193), comment that the distributions of dHint/dt at DON and RVK have a significant component perpendicular to the coast.

  - **This is a good point, this is now mentioned on L215**

- Section 4.2.1: I suggest removing the discussion on how you have achieved the mean direction of dH/dt here. This has been defined in Section 2.2 (Methods section). Move the mention of the Davis (2002) method to section 2.2.

- - **Modifications done as suggested. Section 4.2.1 removed and text added to Section 2.2.**

- L234: Figure 3, panel 4)

  - **OK, L254**

- Section 4.5: The reader is left with the idea that, despite the efforts made in this manuscript, forecasting GIC is still an equally distant undertaking. Do they really want to transmit this notion, perhaps in line with the conclusion of Pulkkinen et al., 2006, that "dBx/dt and dBy/dt fluctuations are not even in principle predictable in a deterministic way"?

  - **Section 4.5 was re-written to include more discussion on simulations and the difficulty of predicting dH/dt and GIC based on Pulkkinen et al. (2006),Amariutei and Ganushkina, 2012), Wintoft et al. (2015), Ngwira et al., 2018; Engebretson et al., 2019a, b)**

  Moreover, please note that forecasting GIC (title) is not equivalent to forecasting dH/dt (first line). Did the authors mean "dH/dt" in the title instead of "GIC"? Also, L151-153 are especially confusing to me. For these reasons, I would recommend either rewriting this section more clearly or consider removing it.

  - **Major additions and modifications were made in section 4.5 to include more discussion on the relation of dH/dt and GIC, and GIC forecasts based on Wintoft et al. (2015).**

**References**

Amariutei, O. A. and Ganushkina, N. Yu.: On the prediction of the auroral westward electrojet index, Ann. Geophys., 30, 841–847, https://doi.org/10.5194/angeo-30-841-2012, 2012.

Engebretson, M. J., Pilipenko, V. A., Ahmed, L. Y., Posch, J. L., Steinmetz, E. S., Moldwin, M. B., Connors, M. G., Weygand, J. M., Mann,I. R., Boteler, D. H., Russell, C. T., and Vorobev, A. V.: Nighttime Magnetic Perturbation Events Observed in Arctic Canada: 1. Survey and Statistical Analysis, Journal of Geophysical Research: Space Physics, 124, 7442–7458, https://doi.org/10.1029/2019JA026794, _eprint: https://agupubs.onlinelibrary.wiley.com/doi/pdf/10.1029/2019JA026794, 2019a.

Engebretson, M. J., Steinmetz, E. S., Posch, J. L., Pilipenko, V. A., Moldwin, M. B., Connors, M. G., Boteler, D. H., Mann, I. R., Hartinger, M. D., Weygand, J. M., Lyons, L. R., Nishimura, Y., Singer, H. J., Ohtani, S., Russell, C. T., Fazakerley, A., and Kistler, L. M.: Nighttime Magnetic Perturbation Events Observed in Arctic Canada: 2. Multiple-Instrument Observations, Journal of Geophysical Research: Space Physics, 124, 7459–7476, https://doi.org/10.1029/2019JA026797, _eprint: https://agupubs.onlinelibrary.wiley.com/doi/pdf/10.1029/2019JA026797, 2019b.

Juusola, L., Kauristie, K., Vanhamäki, H., Aikio, A., and van de Kamp, M. (2016), Comparison of auroral ionospheric and field-aligned currents derived from Swarm and ground magnetic field measurements, J. Geophys. Res. Space Physics, 121, 9256– 9283, doi:10.1002/2016JA022961.

Kwagala, N.G., M. Hesse, T. Moretto, P. Tenfjord, C. Norgren, G. Tóth, T. Gombosi, H.M. Kolstø and S.F. Spinnangr, 2020: Validating the Space Weather Modeling Framework (SWMF) for applications in northern Europe. Ground magnetic perturbation validation. J. Space Weather Space Clim. 2020, 10, 33, doi:10.1051/swsc/2020034.

Wintoft P., M. Wik & A. Viljanen. Solar wind driven empirical forecast models of the time derivative of the ground magnetic field. J. Space Weather Space Clim., 5, A7, 2015, DOI: 10.1051/swsc/2015008.

---

## Editor Decision (ED1)

Figure 2: What about this one (or similar), where d$\Delta$**H**/d$t$ is also depicted?

[Figure]

Ignoring $\Delta$ is a common practice within space physics community, and makes notations a little simpler. We will add a mention of this in the text.

**H** has a definite and widespread meaning within the geomagnetic community: the total horizontal geomagnetic field vector. The **H** used in the manuscript refers to a perturbation, i.e., a difference from a baseline, which is something substantially different. Please, use $\Delta$**H** to avoid confusion. It may even be mentioned in the text that the suppression of $\Delta$ is a common practice in space science to refer to the perturbation, although its use is preferable to conform to the generalized definition of **H**: the total horizontal geomagnetic field vector. Likewise, even if the time derivative of the baseline is small, please use d$\Delta$**H**/d$t$ instead of d**H**/d$t$ to be more precise (I missed it in my first review).

Note that the two occurrences of "d" in d$\Delta$**H**/d$t$ should not be italicized, as they are not variables.

Regarding the reliability of the external/internal separation of the geomagnetic field perturbations, I understand that the arguments given by the authors in the new version of the manuscript and in their response to my comments point to a reasonable separation. However, given the importance of this point in the manuscript, the authors should give further arguments beyond more or less reasonable speculation. I also understand that they do not have an alternative code at hand to compare separation methods independent from each other (based, e.g., on SCHA, or on an EM forward solver capable of providing the separation given the ground conductivity structure shown in Juusola et al., 2020). So, in an attempt of flexibility on my side, I propose to the authors an alternative check based on SECS:

1- Choose (at least) one variable from all the variables that the authors have represented in the manuscript. For example, $\mathrm{d}\Delta\mathbf{H}/\mathrm{d}t$ ext. and $\mathrm{d}\Delta\mathbf{H}/\mathrm{d}t$ int. at SOD station for a certain year (Figure 7 a and b).
2- Run the SECS code with a minimum number of geomagnetic stations in the IMAGE network, e.g., 10. These 10 stations must be selected randomly. This will probably give a directional distribution quite different from the one represented in the mentioned figure.
3- Repeat the process iteratively a significant number of times with 10 stations randomly selected at a time. Compute the mean of all the directional distributions with 10 stations. This will give a mean distribution associated with 10 stations.
4- Now repeat steps 2 and 3 by choosing 11 stations randomly, then 12, and so on until you complete the original network.
5- Think of a quantity that allows to quantify the difference between the final distribution (i.e., with all the stations) and each of the test distributions. For example, the standard deviation of the difference of two directional distributions.
6- Represent (perhaps in another Appendix) the evolution of this quantity as a function of the number of stations used. The resulting curve should converge towards a stable value as the number of stations increases.

Although some objection could be argued about this validation process, it is a reasonable check and, in case the curve does not converge (I hope it does), it would at least provide the reader with an idea of the reliability of the external/internal separation procedure employed in the manuscript.
* * *
After rereading the manuscript, I realize that the significance of Figure 9 is rather limited, and I think it could be improved. The values in this figure are

calculated from directional distributions as those represented in Figure 7. However, some of the distributions are appreciably even (e.g., that of 1998 for the internal contribution of dH/dt), while others are narrower (that of 2008 for the external contribution of dH/dt). Figure 9 provides a single value that does not account for this fact. Moreover, because of the special procedure followed to calculate the mean angle, this mean tends to be shifted towards 180º with respect to the most likely direction. I'm not willing to start a new discussion here on the procedure for calculating the mean; however, I suggest that a parameter be included in Figure 9 that accounts for the "evenness" or uniformity of the distributions in Figures 6 and 7 (thus accounting for the significance of the provided mean angle). This parameter could again be the standard deviation of the distribution, which is larger for the year 1998 than for 2008. This can be represented in Figure 9 as bars superimposed on the mean values.

Minor points:

L39: "the" is repeated.

L48: "more complex than **that** of"

L75: Please, cite the studies the authors refer to.

L78: "… where $\Delta\mathbf{H}$ is the **baseline-subtracted** total …"

Figure 3, panel 5): the authors have selected a high range of values for the vertical axis (-100º to 100º) to highlight the small amplitude of the variations of $\Delta\theta(\Delta\mathbf{H})$ compared to those of $\Delta\theta(d\Delta\mathbf{H}/dt)$. However, the range of this axis does not coincide with the range of panel 6. Either impose the same range, o rather use a more adjusted range, commensurate with the depicted variation, e.g., +-20º.

L166: Add something like: "… 90 degrees**, which is the mean of an even distribution in $\Delta\theta$**".

L192: This citation is inappropriate. The referred article performs the separation based on Spherical Harmonic Analysis (SHA), which is suitable for the entire globe. In any case, perhaps Spherical Cap Harmonic Analysis (SCHA) could be one of the suggested regional methods, but it should be noted that difficulties could arise if the size of the sources is larger than that of the region under study. In fact, many of the workers that attempted to model external field variations by SCHA encountered difficulties in the proper separation of external and internal fields for the above-mentioned reason, as discussed in Torta (2020)*. In the case of high latitudes, the main source of geomagnetic

disturbances is closely related to the auroral electrojet, which is especially limited in latitude, so SCHA could probably perform comparably to SECS.

*Torta, J.M., 2020, Modelling by Spherical Cap Harmonic Analysis: A Literature Review, Surv Geophys 41, 201–247. https://doi.org/10.1007/s10712-019-09576-2*

L193: "... will be a small portion of **the true** external field present in the **modelled** internal field ..."

L252: "The **mean of** the relative ..."

L261 and 162: Replace "Appendix ..." with "Figures ... in Appendices A and B".

L275: Viljanen et al. (2001, p. 1110)

Figure A2: Use the format yyyy/mm instead of yyyym or yyyymm in the headers of each subplot.

---

## Author Response (AR2)

**Angeo-2022-4 Response to referee**

We thank the editor and referee for taking the time to read our revised manuscript. After carefully reading and considering the referee comments, we have made the following revisions (**in bold**) to the manuscript.
* * *
- Figure 2: What about this one (or similar), where d$\Delta$**H**/d*t* is also depicted?
  - **We find this graph useful and have included a similar figure in our manuscript as Fig 2, as suggested.**
- "… Please, use $\Delta$**H** to avoid confusion. … Note that the two occurrences of "d" in d$\Delta$**H**/d*t* should not be italicized"
  - **After considering this point, we decided to change all occurrences, where H refers to the baseline subtracted magnetic field vector, to "$\Delta$H". We also corrected the formatting of d's in the derivative notations.**
- "Regarding the reliability of the external/internal separation of the geomagnetic field perturbations, I understand that the arguments given by the authors in the new version of the manuscript and in their response to my comments point to a reasonable separation. However, given the importance of this point in the manuscript, the authors should give further arguments beyond more or less reasonable speculation. I also understand that they do not have an alternative code at hand to compare separation methods independent from each other (based, e.g., on SCHA, or on an EM forward solver capable of providing the separation given the ground conductivity structure shown in Juusola et al., 2020). So, in an attempt of flexibility on my side, I propose to the authors an alternative check based on SECS: … "
  - **This is a valid suggestion, and actually a similar reliability analysis has been performed previously in Juusola et al., 2020 (Section 4.3):** "… *By removing the three nearest stations of ABK, KIL, and MUO, we can significantly decrease the density of the network around KIR. We run the magnetic field separation with this reduced network and then compute k, similar to the analysis presented in Sect. 3.2 and 3.3. The resulting internal contributions are 26 % (22 %) for Bx, 39 % (30 %) for By, 58 % (47 %) for dBx/dt, and 66 % (51 %) for dBy/dt. The numbers in parentheses give the corresponding contribution for the intact network (Table 1). There is some increase in the internal contribution with the reduced network, indicating that structures smaller than what the network can resolve at 90 km altitude may be mapped underground instead. However, the relative behavior of the different parameters remains unchanged. This indicates that although our numbers are somewhat sensitive to the station configuration, the conclusions drawn from them should still be valid.*"
  - **We also note that since our analysis covers over 20 years, the number of available stations changes over that time. We have included the number of stations used in SECS separation each year in Figure 10 (also attached here). The amount of stations has doubled between 1996 and 2018. Still, there is no obvious trend, for example, in Fig. 9, which shows the mean direction and standard deviation of magnetic field directions at three stations. This suggests that the number of stations used in SECS separation does not significantly affect our results. Mention of this was also added to Section 4.1 L 211-214.**

- **Finally, a thorough analysis on the reliability of the SECS method should be done in the future. However, it is not a straightforward task. Comparing to a different method would only tell about differences between the chosen methods. So far, there is no "ground truth" to which to compare the method. Also, running the SECS code for different station configurations takes time. Repeating the analysis, for example, for one year of data, on ten different station configurations would take over a month of computing time. This reliability check is still an important and interesting topic for future manuscripts.**
- "After rereading the manuscript, I realize that the significance of Figure 9 is rather limited, and I think it could be improved. …"
  - **Figure 9 was modified to include standard deviations for each year. Also relevant text was added L155-159.**

Minor points:
- L39: "the" is repeated.
  - **Fixed**
- L48: "more complex than that of"
  - **OK**
- L75: Please, cite the studies the authors refer to.
  - **Citations to Viljanen, 2001 and Viljanen, et al. 2011 were added L 75**
- L78: "… where $\Delta \mathbf{H}$ is the baseline-subtracted total …"
  - **OK, L 79**
- Figure 3, panel 5): the authors have selected a high range of values for the vertical axis (-100º to 100º) to highlight the small amplitude of the variations of $\Delta\theta(\Delta\mathbf{H})$ compared to those of $\Delta\theta(\mathrm{d}\Delta\mathbf{H}/\mathrm{d}t)$. However, the range of this axis does not coincide with the range of panel 6. Either impose the same range, o rather use a more adjusted range, commensurate with the depicted variation, e.g., +-20º.
  - **Changed the y-range in Fig. 3 panel 5 to the suggested +/- 20º.**
- L166: Add something like: "… 90 degrees, which is the mean of an even
- distribution in $\Delta\boldsymbol{\theta}$".
  - **OK**
- L192: This citation is inappropriate. The referred article performs the separation based on Spherical Harmonic Analysis (SHA), which is suitable for the entire globe. In any case, perhaps Spherical Cap Harmonic Analysis (SCHA) could be one of the suggested regional methods, but it should be noted that difficulties could arise if the size of the sources is larger than that of the region under study. In fact, many of the workers that attempted to model external field variations by SCHA encountered difficulties in the proper separation of external and internal fields for the above-mentioned reason, as discussed in Torta (2020)*. In the case of high latitudes, the main source of geomagneticdisturbances is closely related to the auroral electrojet, which is especially limited in latitude, so SCHA could probably perform comparably to SECS. *Torta, J.M., 2020, Modelling by Spherical Cap Harmonic Analysis: A Literature Review, Surv Geophys 41, 201–247. https://doi.org/10.1007/s10712-019-09576-2
  - **Previous citation removed and replaced with *Torta, 2020*. L202**
- L193: "… will be a small portion of the true external field present in the modelled internal field …"
  - **OK, L202-203**
- L252: "The mean of the relative …"
  - **OK, L264**
- L261 and 162: Replace "Appendix …" with "Figures … in Appendices A and B".
  - **OK, L273-274**
- L275: Viljanen et al. (2001, p. 1110)

- ○ **OK, L288**
- Figure A2: Use the format yyyy/mm instead of yyyym or yyyymm in the headers of each subplot.
  - ○ **Ok, date format changed as suggested in Fig. A2**

[Figure]

*Figure 1: New Figure 10 in manuscript. Number of stations used in SECS analysis each year. Numbers are calculated from daily values.*

---

## Author Response (AR3)

2022-08-02

**Authors' response**

We thank the editor and referee for reading the revised manuscript. We are happy to hear our revisions were sufficient and only minor modifications are needed. The following additions were made to the text:

**The referee's technical comments:**

Text taken from the last "Response to Referee" file:

"This is a valid suggestion, and actually a similar reliability analysis has been performed previously in Juusola et al., 2020 (Section 4.3): "… By removing the three nearest stations of ABK, KIL, and MUO, we can significantly decrease the density of the ..."

Please, mention in the manuscript (section 4.1) that the above reliability analysis has been performed in Juusola et al. (2020), and state its main conclusions in addition to those already stated to justify a reliable external/internal separation.

**As suggested we added the following lines into manuscript Section 4.1, L215-218 :**

*"… Also in Juusola et al. (2020, Section 4.3) the authors performed a simple analysis on the reliability of the SECS separation by decreasing the density of stations used in the analysis. Their main conclusion was that even though there is a small increase in the internal contribution with the reduced network, the relative behavior of the different parameters is unchanged. ..."*